# Thioester synthesis through geoelectrochemical $CO_2$ fixation on Ni sulfides

Norio Kitadai [1,2 ✉], Ryuhei Nakamura[2,3], Masahiro Yamamoto[1], Satoshi Okada [1], Wataru Takahagi [1,4], Yuko Nakano[2], Yoshio Takahashi[5], Ken Takai [1] & Yoshi Oono[6]

A prevailing scenario of the origin of life postulates thioesters as key intermediates in protometabolism, but there is no experimental support for the prebiotic $CO_2$ fixation routes to thioesters. Here we demonstrate that, under a simulated geoelectrochemical condition in primordial ocean hydrothermal systems (–0.6 to –1.0 V versus the standard hydrogen electrode), nickel sulfide (NiS) gradually reduces to $Ni^0$, while accumulating surface-bound carbon monoxide (CO) due to $CO_2$ electroreduction. The resultant partially reduced NiS realizes thioester (S-methyl thioacetate) formation from CO and methanethiol even at room temperature and neutral pH with the yield up to 35% based on CO. This thioester formation is not inhibited, or even improved, by 50:50 coprecipitation of NiS with FeS or CoS (the maximum yields; 27 or 56%, respectively). Such a simple thioester synthesis likely occurred in Hadean deep-sea vent environments, setting a stage for the autotrophic origin of life.

[1] Super-cutting-edge Grand and Advanced Research (SUGAR) Program, Institute for Extra-cutting-edge Science and Technology Avant-garde Research (X-star), Japan Agency for Marine-Earth Science and Technology (JAMSTEC), Yokosuka, Japan. [2] Earth-Life Science Institute, Tokyo Institute of Technology, Meguroku, Tokyo, Japan. [3] Biofunctional Catalyst Research Team, RIKEN Center for Sustainable Resource Science, Wako, Saitama, Japan. [4] Department of Chemistry, Graduate School of Science, The University of Tokyo, Bunkyo-ku, Tokyo, Japan. [5] Department of Earth and Planetary Science, Graduate School of Science, The University of Tokyo, Bunkyo-ku, Tokyo, Japan. [6] Department of Physics, University of Illinois at Urbana-Champaign, Urbana, IL, USA. ✉email: nkitadai@jamstec.go.jp

Thioesters occupy the central position in cellular metabolism as exemplified by acetyl-CoA. This fact has suggested thioesters to be the crucial components of prebiotic precursor of metabolism since as early as 1960's[1–3]. A key enzyme in thioester biosynthesis is carbon monoxide dehydrogenase/acetyl-CoA synthase (CODH/ACS), in which the Ni-based active centers reduce $CO_2$ to CO, react CO with a methyl group to form acetyl, and acetylate CoA to form acetyl-CoA. This apparently simple process with pivotal Ni has evoked the idea that a prebiotic $CO_2$ fixation on a (Fe,Ni)S mineral preceded the CODH/ACS-based reaction[4,5].

However, experimental support of this scenario remains limited. The $CO_2$-to-CO conversion is endergonic even with $H_2$ as the reductant (Supplementary Fig. 1), so biological CO production uses either electron bifurcation or a chemiosmotic pH gradient across the cell membrane to overcome the energy shortage. In the ACS catalytic cycle, the active Ni site serves as both the electron donor and accepter through changing its oxidation states between +1 and +3, facilitating both the reduction and oxidation intermediate steps[6]. Such redox bifunctionality of Ni has never been observed in natural Ni-bearing minerals. Although Huber and Wächtershäuser demonstrated S-methyl thioacetate (MTA) synthesis from CO and methanethiol in the presence of NiS as a potential prebiotic precursor of the ACS reaction[7], the yield was very low (0.2% based on the initial amount of CO) even under their optimum condition (pH 1.6 and 100 °C), which is not so common in nature. A recent work suggested a one-pot pyruvate synthesis from $CO_2$ and $H_2$ on magnetite ($Fe_3O_4$) and awaruite ($Ni_3Fe$) as a protometabolic acetyl-CoA pathway[8]. The proposed reaction, however, provides no explanation for the origin and antiquity of thioester-dependent metabolism. Thioesters might have played central roles in protometabolism not only as carbon sources but also as energy currencies in a manner analogous to ATP[1,2,9], serving as an entry point of phosphate into metabolism[5,10].

How were $CO_2$-to-CO reduction and CO conversion realized under primordially realistic conditions? A clue lies in an *on-site* observation by Yamamoto et al.[11] of electricity generation in deep-sea hydrothermal vent chimneys and mineral deposits. The geoelectricity arises from the redox coupling between hydrothermal fluid chemicals and seawater-dissolved species via electrically conducting yet thermally insulating sulfide rocks (Supplementary Fig. 1)[12,13]. Considering cool (0–50 °C) and slightly acidic (pH 6–7) character of the ancient seawater[14] together with pH and temperature dependences of the $H^+/H_2$ and $CO_2/CO$ redox potentials, $H_2$-rich alkaline hydrothermal systems must have readily provided negative electric potentials favorable for the $CO_2$-to-CO reduction at the chimney-ancient seawater interface. We previously demonstrated efficient $CO_2$ electroreduction to CO on certain metal sulfide catalysts (for example, cadmium sulfide) under a simulated early ocean geoelectrochemical condition[15]. It was later found that FeS undergoes day-scale electroreduction to $Fe^0$ [16]. The resultant FeS-$Fe^0$ assemblage, named FeS_PERM (FeS partially electroreduced to metal), showed the exceptional capability of promoting various prebiotically important reactions owing to the synergy between surface Fe sites with different oxidation states ($Fe^{2+}$ and $Fe^0$)[17,18].

In our previous works[15,16], NiS exhibited neither CO evolution nor $Ni^0$ formation detectable by X-ray diffraction (XRD) analysis. However, our further investigation presented below found noncrystalline growth of $Ni^0$ under geochemically feasible potential conditions, with a substantial amount of CO bound on the resultant surface $Ni^0$ sites. We will also report here that the partially reduced NiS, that is, NiS_PERM realizes efficient conversion of CO and methanethiol to MTA even at room temperature and neutral pH. Notice that the accumulation of surface-

bound CO during $CO_2$ electroreduction is generally recognized as poisoning of surface (electro)catalytic activity[19,20], but our results suggest that the CO accumulation process on NiS_PERM was rather a crucial step for subsequent primordial thioester formation in early ocean hydrothermal systems. Thioesters are versatile compounds in organic chemical synthesis as well as in biosynthesis, enabling diverse coupling reactions, including C–C bond formation, esterification, and amide bond formation owing to the activated acyl unit[21–23]. To the best of our knowledge, our study provides the first experimental demonstration of thioester synthesis from CO and thiol under mild aqueous conditions. Thus, besides rendering a realistic support for autotrophic scenarios of the origin of life[1–5], our demonstrated CO-thiol reaction on NiS_PERM could be a practical approach to CO utilization[24]. In the following and in Supplementary Information, we also present the results for FeS and CoS and their influences on the NiS's capabilities for CO production and MTA synthesis.

## Results

**Metal sulfide electroreduction**. We prepared metal sulfides, including NiS by simply mixing an aqueous solution of the corresponding metal chloride and an aqueous solution of sodium sulfide. The obtained sulfides were exposed to a constant electric potential (–0.5 to –1.0 $V_{SHE}$ (volt versus the standard hydrogen electrode)) for 7 days in 100 mM NaCl at room temperature (25 ± 2 °C) (Supplementary Figs. 2 and 3). The solution pH was maintained at 6 ± 0.25 by continuous $CO_2$ bubbling. The electrolyzed sulfides were then separated from the supernatant solution, dried under vacuum, and used for the subsequent experiments.

The NiS consists of aggregated nanoparticles with an average particle diameter of 15 ± 11 nm (Supplementary Fig. 6). Although no significant morphological change was discerned (Supplementary Fig. 6), exposure at –0.5 $V_{SHE}$ led to the desulfurization of NiS to heazlewoodite ($Ni_3S_2$) (Fig. 1a), consistent with thermodynamic calculation (Fig. 1b). Further reduction of $Ni_3S_2$ at lower electric potentials was indicated by energy-dispersive X-ray spectroscopy (EDS) mapping on the NiS particles electrolyzed at –1.0 $V_{SHE}$ (Fig. 1c), where a clear decrease in the sulfur signal intensity relative to nickel was observed in comparison with pure NiS, and with the NiS electrolyzed at –0.5 $V_{SHE}$ (Fig. 1d; all EDS data are scaled to have the same S signal intensities at 2.3 keV). In agreement with the EDS result, enhancement of the reducing potential from –0.5 to –1.0 $V_{SHE}$ led spectral changes in the nickel K-edge X-ray absorption near-edge structure (XANES) of NiS samples (Fig. 1e) attributable to the occurrence and growth of $Ni^0$ in $Ni_3S_2$ up to the $Ni^0$ percentage of 12% (Fig. 1f and Supplementary Fig. 11). Similarly, XANES analysis showed 18% conversion of CoS to $Co^0$ at –1.0 $V_{SHE}$ (Supplementary Fig. 12). Thus, NiS_PERM and CoS_PERM are formed at –1.0 $V_{SHE}$ and at even less negative potentials (i.e., closer to 0 $V_{SHE}$) near their sulfide/metal equilibria (Fig. 1b, f, Supplementary Figs. 11 and 12) just as in the FeS case. However, in contrast to FeS_PERM that exhibited broad but clear XRD signals for $Fe^0$ even at –0.7 $V_{SHE}$ (Supplementary Fig. 8)[16], NiS_PERM showed no XRD signal for $Ni^0$ in the examined potential range (≤ –1.0 $V_{SHE}$) (Fig. 1a). The $Ni^0$ percentage up to 12% (Fig. 1f) should be well above the detection limit of XRD (Supplementary Fig. 10) if $Ni^0$ formed a localized crystalline domain in the $Ni_3S_2$ structure. Thus, NiS_PERM would be with more finely dispersed zerovalent metal than FeS_PERM.

**CO production and accumulation on the metal-sulfide PERMs**. To explore the possibility of CO production and accumulation on the metal-sulfide PERMs formed under $CO_2$ atmosphere, we

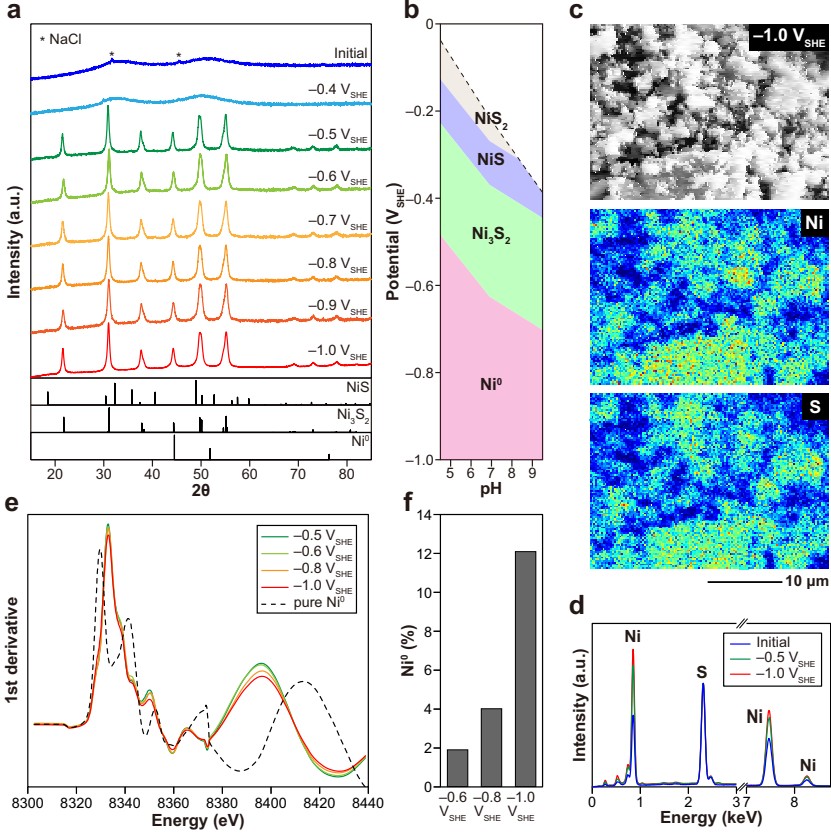

**Fig. 1 Characteristics of the electrolyzed NiS. a** XRD patterns showing NiS electroreduction to $Ni_3S_2$ at −0.5 $V_{SHE}$ or below. **b** $E_H$-pH diagram for the sulfide/metal system of Ni. **c** EDX mapping results of the NiS electrolyzed at −1.0 $V_{SHE}$. **d** Normalized EDX spectra of NiS before (blue) and after the electrolysis at −0.5 $V_{SHE}$ (green) or −1.0 $V_{SHE}$ (red). **e** First derivative nickel K-edge XANES spectra. **f** The percentages of $Ni^0$ in the electrolyzed NiS quantified by the XANES analysis (see "Methods").

dissolved the resultant metal-sulfide PERMs completely in 35% hydrogen chloride, and quantified the released CO by gas chromatography (Supplementary Fig. 18 and Supplementary Movie 1). Except for FeS, all the sulfides exposed to ≤ −0.6 $V_{SHE}$ were found to retain CO (Fig. 2a and Supplementary Table 2). The threshold potential is close to the sulfide/metal equilibrium potentials for Ni and Co (−0.57 $V_{SHE}$ for both; Fig. 1b and Supplementary Fig. 4), and is also near the thermodynamic $CO_2$/CO redox potential at the condition for sulfide electrolysis (−0.50 $V_{SHE}$; Supplementary Fig. 1). The amount of CO increased with decreasing the potential up to $180 \pm 40 \, \mu mol \, g^{-1}$ for the case of NiS. Because sulfurized Ni binds CO considerably more weakly than pure $Ni^{0,25}$ the observed CO should be mostly due to the CO bound on the surface $Ni^0$ sites. In fact, the maximum adsorption ($180 \pm 40 \, \mu mol \, g^{-1}$) corresponds to the surface coverage of one CO molecule per $8 \pm 3$ surface Ni atoms (Supplementary Fig. 13), consistent with the percentage of $Ni^0$ grown at −1.0 $V_{SHE}$ (Fig. 1f). The similar interpretation should be applicable to the adsorption behavior of CO on the CoS_PERM[26]. Coprecipitation with FeS led to decline of the amounts of CO on NiS and CoS (Fig. 2a) probably because the resultant surface Fe sites were not involved in the CO accumulation owing to the stable $Ni^0$–CO and $Co^0$–CO bindings, as indicated by no CO evolution during the NiS and CoS electrolysis[15].

**Nonenzymatic thioester synthesis.** We mixed the metal-sulfide PERMs retaining the surface-bound CO (Fig. 2a) (50 mg for each) with an aqueous solution of sodium methanethiolate (75 μmol) in a serum bottle (13.8 ml), and agitated it at room temperature (25 ± 2 °C) for up to 7 days without externally imposing electric

potential. pH was buffered at neutral (7.0 ± 0.5) by filling the gas space with $CO_2$ (1 atm). Despite this very mild condition, gas chromatography–mass spectrometry analysis revealed efficient MTA formation in the presence of specific electrolyzed sulfides (Fig. 2b, c). Organosulfur compounds detected with the amount higher than 0.01 μmol were methanethiol, MTA, dimethyl sulfide, and dimethyl disulfide (Supplementary Figs. 19 and 20), among which MTA was the sole product built from CO and methanethiol:

$$CO + 2CH_3SH \rightarrow CH_3COSCH_3 + H_2S \quad (1)$$

Experiments with the NiS_PERM prepared at −1.0 $V_{SHE}$ exhibited the following product characteristics. The yield of MTA increased over one week, while the amounts of CO and methanethiol decreased from their maxima at the beginning (Supplementary Fig. 24). CO appeared in the gas-phase through competitive adsorption with methanethiol: in the absence of methanethiol under otherwise identical condition, only a small amount of CO gas was detected (0.03 μmol) and no MTA was observed (Supplementary Fig. 21a). As long as methanethiol was available, replacement of $CO_2$ in the gas space with helium (He), and dissolution of 1 M phosphate (pH 7.0) into the sample solution had no significant influence on the yield of MTA (3.2 ± 0.6 μmol; Fig. 2b → 3.6 ± 0.7 μmol). It was also confirmed with $^{13}C$-labeled $CO_2$ that the $CO_2$ filled in the serum bottle did not serve as a component of MTA. The MTA formed under $^{13}CO_2$ atmosphere showed the mass spectrum identical to that of the standard MTA with normal carbon isotopic composition.

In aqueous solutions, acetate (0.19 ± 0.04 μmol = 0.25 ± 0.05 mM in 0.75 ml $H_2O$) and formate (0.03 ± 0.01 μmol = 0.035 ± 0.01 mM

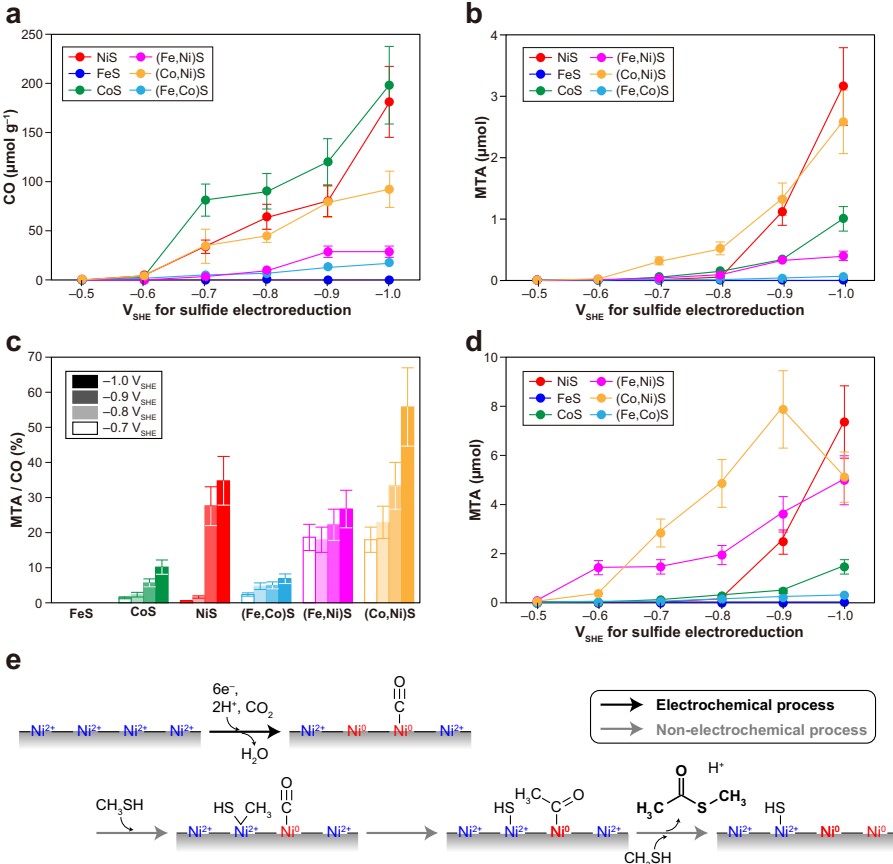

**Fig. 2 CO₂-to-CO electroreduction and CO conversion to S-methyl thioacetate (MTA) on the metal-sulfide_PERMs. a** The amounts of surface-bound CO on the electrolyzed sulfides ($\mu$mol g$^{-1}$). **b** The yields of MTA from the surface-bound CO (**a**) in the presence of methanethiol (75 $\mu$mol per 50 mg of sulfide). **c** The percentages of CO converted to MTA in the reactions shown in (**b**). **d** The yields of MTA from externally added CO and methanethiol (75 $\mu$mol for each). The MTA yield on (Co,Ni)S declined at the lowest potential (–1.0 V$_{SHE}$) because of the depletion of methanethiol (see Supplementary Table 4). **e** Possible intermediate steps in the MTA synthesis via the formation of NiS_PERM with the surface-bound CO (red arrow) and the CO-methanethiol reaction on NiS_PERM (green arrows). The error bars in **a**–**d** were determined by multiple independent runs under the same reaction conditions (see "Methods").

in 0.75 ml H₂O) were observed (Supplementary Fig. 25). The acetate concentration increased to 2.0 ± 0.4 mM when the product solution was basified by 1 M sodium hydroxide due to MTA hydrolysis. Because MTA hydrolysis is accelerated at elevated temperatures[27], a one-day experiment at 80 °C resulted in a higher yield of acetate (2.4 ± 0.5 $\mu$mol = 3.2 ± 0.6 mM in 0.75 ml H₂O) and a lower yield of MTA (0.68 ± 0.14 $\mu$mol) (Supplementary Fig. 25). Other biologically relevant organic acids, such as pyruvate and lactate, were not detected in any product solutions with concentrations higher than 0.01 mM. We also treated solid samples with 8 M potassium hydroxide after the reaction, but neither pyruvate nor lactate was observed.

When CO and methanethiol were initially introduced with equal amounts (75 $\mu$mol, or 0.14 bar in the gas space) in the presence of NiS_PERM formed at –1.0 V$_{SHE}$, the yield of MTA more than doubled, compared with the case with methanethiol alone in the initial gas-phase (3.2 ± 0.6 → 7.4 ± 1.5 $\mu$mol; Fig. 2d and Supplementary Fig. 21a). In contrast, under the same initial condition with CO, few or no detectable MTA was formed in the presence of pure NiS, the Ni₃S₂ prepared at –0.5 V$_{SHE}$, or pure Ni⁰ (Supplementary Fig. 21 and Supplementary Table 4), indicating the necessity of the coexistence of zerovalent and non-zerovalent Ni surface sites for efficient MTA production. It is also notable that the MTA yield decreased steeply as the NiS electroreduction potential was changed from –1.0 to –0.8 V$_{SHE}$

(Fig. 2b, d) although –0.8 V$_{SHE}$ is still sufficient for the Ni⁰ generation (Fig. 1f). In light of reported chemical functions of Ni and other metals in organometallic analogues of ACS[28,29], one possible interpretation is that the MTA synthesis involves at least two adjacent surface Ni⁰ sites (Fig. 2e): one Ni⁰ serves as an electron donor for the reductive cleavage of methanethiol C–S bond. Close location of another Ni⁰ (the closest Ni–Ni distance = 2.48 Å; Supplementary Fig. 13) allows transfer of the generated electrophilic methyl onto the nucleophilic carbon of adsorbed CO. After the formation of C–C bond to make the acetyl-Ni⁰, its oxidative thiolysis to form MTA occurs through coupling with the reduction of neighboring non-zelovalent Ni atoms. The cleavage of methanethiol C–S bond also accounts for the formations of CH₄ and dimethyl sulfide (Supplementary Tables 3 and 4), where the resultant methyl converts to CH₄ via protonation, while thiolysis of the methyl generates dimethyl sulfide. If a random Ni⁰ distribution is assumed, a 12% conversion of Ni²⁺ to Ni⁰ achieved at –1.0 V$_{SHE}$ (Fig. 1f) results in 1.4% occurrence of the adjacent Ni⁰–Ni⁰ surface pair, while this percentage drops to 0.16% with a 4% conversion achieved at –0.8 V$_{SHE}$. Thus, our proposed process taking multiple Ni sites into account (Fig. 2e) explains the large potential dependence in MTA productivity of NiS_PERM (Fig. 2b, d) as well as the few or no MTA formation on Ni₃S₂ and pure Ni⁰. This nonenzymatic process is expected to be eventually deactivated due to cumulative

surface sulfurization (Fig. 2e), but the surface activity may be restored by additional solid electrolysis.

Interestingly, even greater CO-to-MTA conversion ratios were obtained with the NiS coprecipitating with FeS or CoS (Fig. 2c). Up to $56 \pm 10\%$ of the surface-bound CO, produced by $CO_2$ electroreduction, was converted to MTA on the electrolyzed (Co, Ni)S. Although the electrolyzed (Fe,Ni)S produced MTA with low yields (Fig. 2b) due to the low surface accumulation of CO during the electroreduction, the percentages of CO to form MTA were kept at high levels ($\sim20\%$) even at $\geq -0.8\ V_{SHE}$ (Fig. 2c). In fact, (Fe,Ni)S produced a considerable amount of MTA ($1.44 \pm 0.29$ $\mu$mol) from externally introduced CO even when $-0.6\ V_{SHE}$ was applied to the electrolysis (Fig. 2d). Since the electrochemical reducibility of FeS is higher than that of NiS (Supplementary Figs. 7 and 8), it may well be the case that Fe in the electrolyzed (Fe,Ni)S assists the performance of Ni as an electron donor at an intermediate reduction step just as the supportive role of Fe in ferredoxin serving as the redox mediator in the ACS-reaction[6]. Co is also involved in the enzymatic process as a transporter of the methyl group[6].

## Discussion

Our experiments revealed that partial electroreduction product of NiS, that is, NiS_PERM, promotes thioester synthesis from CO and methanethiol. NiS_PERM accumulates surface-bound CO during the formation under $CO_2$ atmosphere, so NiS offers a two-step $CO_2$ fixation route to thioester with the aid of electric energy. The electric potential required for these reactions ($\leq -0.6\ V_{SHE}$; Figs. 1 and 2) is more reducing than that observed in a present-day black smoker system ($\geq -0.022\ V_{SHE}$)[11]. However, the value of $-0.6\ V_{SHE}$ and even lower potentials are available in $H_2$-rich alkaline hydrothermal fields (Supplementary Fig. 1)[30,31]. Deep-sea systems are more advantageous than terrestrial ones because the elevated pressure increases the solubility of $H_2$, which indeed serves as an electron source for the reduction of seawater $CO_2$ in the presence of (Fe,Ni)S precipitates[32]. On the primordial ocean floor, widespread occurrence of elevated hydrothermal $H_2$ fluxes has been suggested from the presence of awaruite in many fossilized serpentinization systems[33,34].

In the Archean to Hadean eons, submarine hydrothermal environments may also have favored NiS precipitation owing to the huge supply of mantle-derived Ni into the ocean. The resultant massive Ni sulfide deposits are seen today in association with Archean komatiite with the Ni/Fe weight ratios occasionally exceeding one[35]. Given the estimated $Ni^{2+}$ and $Fe^{2+}$ concentrations in the early Archean seawater with their sulfide solubilities, selective NiS precipitation relative to FeS is expected in micro- to semimicro-molal level presence of hydrogen sulfide ($H_2S$ and $HS^-$) (Supplementary Fig. 26), which is a likely concentration range at the outer surface of ancient alkaline hydrothermal chimneys[36].

Within deep-sea vent chimneys, dynamic mixing of hydrothermal fluids with seawater through the pore systems is envisioned (Fig. 3)[37]. Occasional increase of hydrogen sulfide concentration would remove the surface-bound CO (see "Methods", Supplementary Figs. 27 and 28). On the other hand, such $H_2S$-rich condition is advantageous to thiol synthesis from CO or $CO_2$.[38,39] Once $H_2S$ recedes, methanethiol may react with restored CO to form MTA on Ni sulfide_PERMs (Fig. 2b–d). Thus, fluid-mediated material transport should have favored the occurrence and combination of multiple reactions through interlinked mineral pores (Fig. 3). It is conceivable, for example, that the CO formed on NiS_PERM and CoS_PERM is trapped on (Fe,Ni)S_PERM with thiols, thereby realizing the thioester formation with mild electric potentials (Fig. 2d). Thus, the thioester

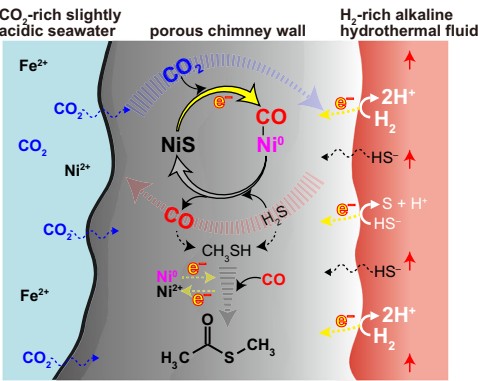

**Fig. 3 Schematic cross-section of a vent chimney in a primordial ocean alkaline hydrothermal system showing possible abiotic thioester synthesis promoted by Ni sulfide_PERM.** Note that the availability of methanethiol in primordial deep-sea vent environments remains controversial and needs further investigation[38,39,58].

synthesis via the formation of Ni sulfide_PERMs should have been possible in primordial ocean alkaline hydrothermal systems. Just as Huber and Wächtershäuser[7], however, we have not yet discussed seriously the source of methanethiol. To make a realistic estimate for the activity and robustness of abiotic thioester synthesis, further investigation for the geochemical availabilities of thiols and the other necessary reaction parameters (the electric potential and the Ni-sulfide composition) is desirable.

Nowadays, growing consensus for $CO_2$ as the dominant carbon species on the Hadean Earth has stimulated exploration of geo- and astro-chemical events that could have transiently realized highly reducing early atmosphere[40]. Subsequent photochemical and aqueous-phase processes with reactive carbon sources, powerful oxidizing/reducing agents, and/or UV light might have realized abiotic thioester synthesis[41,42]. Although these reactions might have played a role in enriching prebiotic soup, their relevance to biosynthesis is ambiguous. Consequently, numerous questions remain unanswered by the transient scenario as to the gap between prebiotic chemistry and early biochemistry conserved in the metabolic and phylogenetic architectures of life.

The two key facts: $CO_2$ fixation utilizing various states of Ni due to our experiments (Fig. 2) and the well-known central role of Ni at the active sites of CODH/ACS[6] may lead to a plausible scenario for the emergence of protometabolism prior to the origin of life. Sustained occurrence of electrochemical energy adequate for the Ni sulfide_PERM formation, $\leq -0.6\ V_{SHE}$ in the ancient seawater condition (Figs. 1 and 2), is limited in nature except for deep-sea hydrothermal settings (Supplementary Fig. 1). This environmental specificity with the known necessity of Ni in both biological and chemical $CO_2$ fixation suggest that the Ni sulfide_PERM-promoted thioester synthesis in early ocean hydrothermal systems (Fig. 3) is a primordial precursor of the CODH/ACS reaction. The CO production on metal-sulfide_PERMs (Fig. 2a) may also have served as crucial carbon and energy sources for the early chemolithotrophic ecosystem, thereby supporting, or even directing, the origin and early evolution of autotrophs as suggested from the autotrophic metabolism core[43].

## Methods

**Preparation of metal sulfides.** All metal sulfides were prepared by adding 100 mM sodium sulfide ($Na_2S$) dropwisely into the corresponding 100 mM metal chlorides ($NiCl_2$, $FeCl_2$, or $CoCl_2$) or their binary mixtures (for example, 50 mM $NiCl_2$ plus 50 mM $FeCl_2$) under vigorous stirring to a final volume ratio of 1:1. Solid precipitates were then separated from the supernatant solutions by centrifugation (8000 rpm, 10 min) and were dried under vacuum. To prevent

oxidation by atmospheric $O_2$, the sample preparation was conducted in a glove box filled with $N_2$ gas (>99.99995% purity), with 4% $H_2$ (>99.99999% purity) being added (the Coy system). All chemicals were purchased from FUJIFILM Wako Pure Chemical Corporation as reagent grade. Deaerated Milli-Q water (18.2 megohms) was used as the solvent.

Pure metals ($Fe^0$, $Co^0$, and $Ni^0$) were obtained from EM Japan. Their reported characteristics and XRD patterns measured in this study are presented in Supplementary Table 1 and Supplementary Fig. 9, respectively.

**Sulfide electrolysis**. Metal sulfide electrolysis was conducted under a simulated early ocean condition in accordance with the procedure reported previously[16]. Briefly, 400 mg of sulfide samples was deposited on a carbon working electrode (5.7 cm$^2$) in a H-type cell (Supplementary Fig. 2), immersed in a deaerated 100 mM NaCl, and exposed to a flow (20 ml min$^{-1}$) of $CO_2$ (>99.995% purity) containing 4 ppm of $H_2S$. The $CO_2$ gas buffered the solution pH at slightly acidic (6.0 ± 0.25). The $H_2S$ partial pressure was determined by thermodynamic calculation to supply $H_2S$ and $HS^-$ into solution with the equilibrium total concentration of 0.5 μmol kg$^{-1}$. Although the ionic strength and pressure conditions adopted in this experiment are different from the ancient deep-sea hydrothermal settings[36,44], a thermodynamic calculation indicates that these differences have no significant influence on the redox potentials for the sulfide/metal and the $CO_2$/CO systems (Supplementary Fig. 5)[16].

While keeping the $CO_2$ gas flow that was started at least 1 h before each experiment, a constant potential was applied on the carbon electrode for 7 days by using a multi-potentiostat (PS-08; Toho Technical Research). All potentials were measured against an Ag/AgCl reference electrode in saturated KCl and were converted to the SHE scale by the following equation:

$$E(\text{versus SHE}) = E(\text{versus Ag/AgCl}) + 0.198V \tag{2}$$

After the electrolysis, the electrochemical cell was immediately transferred into a glove box filled with $N_2$ and $H_2$ gases (volume ratio, 96:4). The solid sample was then separated from the supernatant solution, dried under vacuum for 1 h, and stored inside the glove box. A dry vacuum pump (RDA-281H, ULVAC) with the reported ultimate pressure of 0.08 Pa was used for the dehydration. The surface-bound CO was expected to be intact at this vacuum condition[25,45,46].

The prepared sulfide_PERMs were stable in the anaerobic glove box over weeks, but were oxidized rapidly in the air. Typically, a minute exposure to the air completely degraded their capabilities for promoting MTA synthesis. Their high susceptibilities to oxidation are not a severe problem in the geoelectrochemical scenario (Supplementary Fig. 1) because of the absence of $O_2$ and other reactive oxidants (for example, $H_2O_2$) in primordial deep-sea environments.

**Solid characterization**. Scanning electron microscopy (SEM) imaging was performed on a Helios G4 UX (Thermo Fisher Scientific) equipped with a PP-3010 cryo preparation system (Quorum) and an Octane Super C5 EDS detector (AMTEC). An acceleration voltage of 2 kV was applied for secondary electron imaging and 20 kV for EDS analysis under a reduced pressure of $<1 \times 10^{-4}$ Pa at room temperature (23 ± 2 °C). Samples were prepared as follows. First, sulfide sample sealed in a serum bottle was placed in a vacuum chamber of the cryo-preparation system, whose internal atmosphere was replaced with $N_2$ gas from liquid $N_2$ vaporization by three times of evacuation-purge cycles. Under flow of $N_2$ gas, the serum bottle was opened, and the sulfide was rubbed onto a carbon tape (Nisshin EM) attached on a transfer shuttle by a plastic spatula. The shuttle was transferred into a transfer device under ~$10^2$ Pa, and then to the SEM chamber via a cryo-preparation chamber (<$10^{-4}$ Pa).

Particle size distribution (Supplementary Fig. 6) was determined by manual segmentation on Affinity Photo for iPad 1.6.8.77 (Serif (Europe) Ltd.) followed by image analysis with Image-Pro 3D v9.3 (Media Cybernetics). Particle diameters were calculated from the segmented areas assuming spherical morphology.

X-ray diffraction (XRD) patterns of sulfide samples were measured by using an X-ray diffractometer with Cu Kα radiation (MiniFlex 600, Rigaku). All runs were conducted with 2θ ranging from 10° to 90° using 0.02° 2θ step with a scan rate of 0.1 or 1° min$^{-1}$. To prevent oxidation by atmospheric $O_2$ during the measurement, the solid samples were shielded in an air-sensitive sample holder (Rigaku). Peak identifications were made on the basis of the reference patterns reported in the Powder Diffraction File published by the International Centre for Diffraction Data. The reference patterns are presented in Fig. 1a and Supplementary Figs. 8–11 with the measured XRD data.

X-ray absorption near-edge structure (XANES) spectra at Ni and Co K-edges were measured at BL-12C (bending magnet beamline) in a synchrotron radiation facility (Photon Factory) in High Energy Accelerator Research Organization (KEK), Tsukuba, Japan. In the beamline, X-ray from a synchrotron operated at 2.5 GeV (current: 450 mA) was monochromatized with a Si(111) double-crystal monochromator, and focused to an area of $0.5 \times 0.5$ mm$^2$ with a bent cylindrical mirror, which also reduced the higher order. XANES spectra were obtained in transmission mode using two ion chambers to measure intensities of incident ($I_0$) and transmitted (I) X-rays. Energy step within the XANES region was 0.25 eV, and the absorbance by the sample (μt) was obtained as μt = ln(I/$I_0$). Sulfide samples were diluted with boron nitride (BN, < 150 nm, 99% purity) to yield the metal/BN molar ratio of 1:25, and shielded by an $O_2$-impermeable polyethylene film.

Measurement was conducted for a part of the sample with uniform thickness within the area of the X-ray beam.

The percentages of zerovalent metal in the electrolyzed NiS and CoS were estimated by a least-squares fitting of the sample spectra after background-subtraction and normalization with an X-ray absorption spectroscopy data processing software ATHENA[47]. For the NiS samples, the linear combination of pure $Ni^0$ and the NiS electrolyzed at −0.5 $V_{SHE}$ were used for the fitting between 8300 and 8440 eV. The NiS electrolyzed at −0.5 $V_{SHE}$ was used as a reference of $Ni_3S_2$ based on the thermodynamic calculation (Fig. 1b), XRD pattern (Fig. 1e), and the XANES spectral profile (Supplementary Fig. 11a) identical to that of pure $Ni_3S_2$.[48] The energy region higher than 8440 eV was not analyzed because of lower signal-to-noise ratio and of greater influence from the local environment of Ni atom. The best fit was determined by calculating the lowest R factor, which was defined as:

$$R = \sum \left( \mu_{\exp}(E) - \mu_{cal}(E) \right)^2 \tag{3}$$

In this equation, $\mu_{\exp}$ and $\mu_{cal}$ are the experimental and the calculated absorbances at a given energy $E$, respectively. For the CoS samples, the energy range 7690–7830 eV were fitted with the linear combination of pure $Co^0$ and the CoS electrolyzed at −0.5 $V_{SHE}$ in a manner similar as Ni described above.

**Search for the surface-bound CO on electrolyzed sulfides**. Each sulfide sample (50 mg for each) was sealed in a serum bottle (124.7 ml) with a Teflon-laminated butyl rubber cap and an aluminum stopper in a glove box filled with $N_2$ and $H_2$ gases (volume ratio, 96:4). The bottle was then filled with pure He gas (>99.99995%) by flowing the He through a stainless needle at a rate of 100 ml min$^{-1}$ over 10 min. This was followed by the addition of 5 ml of 35% HCl (super special grade; FUJIFILM) under vigorous agitation. After the sulfide sample dissolved completely under 60 rpm rotation of the bottle at room temperature (25 ± 2 °C) (typically, within 30 min), the headspace gas was analyzed by gas chromatography (GC) (Fig. 2a, Supplementary Fig. 18 and Supplementary Table 2). To ensure reproducibility, we carried out multiple independent runs, using several sulfide samples for each metal. Differences among the data obtained at the identical condition were less than 20%. The same was true for the following two experiments "Nonenzymatic thioester synthesis" and "Competitive adsorption of CO with $H_2S$".

To verify the CO production and accumulation on the NiS_PERM, we conducted the following experiment with isotopically labeled sodium bicarbonate (NaH$^{13}$CO$_3$, 99% purity) provided by Cambridge Isotope Laboratories, Inc. First, 400 mg of NiS was electrolyzed at −1.0 $V_{SHE}$ for 7 days in 60 ml of 0.5 M phosphate buffer solution (pH 6.0) under pure $N_2$ gas flow (>99.9998%). Then, the gas headspace of the electrochemical cell (~40 ml) was purged with $^{13}CO_2$ gas that was provided by a dropwise addition of 0.1 M phosphoric acid ($H_3PO_4$) onto NaH$^{13}$CO$_3$ powder under vigorous stirring in a closed vial connected with the electrochemical cell with a polyvinyl chloride tube and stainless needles. After additional two-hour electrolysis with the gas inlet and outlet closed, the NiS_PERM on the carbon paper was collected, dried under vacuum, and dissolved completely in 35% HCl. GC-MS analysis of the released gas revealed the formation of $^{13}CO$ with the $m/z = 29$ (Supplementary Fig. 18c). No $^{13}CO$ was detected in the $^{13}CO_2$ formed by the $H_3PO_4$ addition to NaH$^{13}$CO$_3$ (Supplementary Fig. 18).

In the normal $CO_2$ gas bubbled during the sulfide electrolysis, we detected no CO with our GC's maximum performance (detection limit; ~1 ppm). Even if a very slight amount of CO impurity is present in the $CO_2$, its contribution to the surface-bound CO (Fig. 2a and Supplementary Table 2) must be negligible because the electrolyzed FeS released almost no CO despite the fact that $Fe^0$ has an even stronger CO binding energy than $Ni^0$ and $Co^0$.[49]

Supplementary Movie 1 presents a demonstration of the surface-bound CO on the NiS electrolyzed at −1.0 $V_{SHE}$. A weak acid (1 M $H_3PO_4$) was used here to make sure that the gas bubble formation is not due to NiS dissolution but from the purging of the electrolyzed surface.

**Nonenzymatic thioester synthesis**. Each sulfide sample (50 mg for each) was sealed in a serum bottle (13.8 ml) with a Teflon-laminated butyl rubber cap and an aluminum stopper in a glove box filled with $N_2$ and $H_2$ gases (volume ratio, 96:4). The bottle was then filled with pure $CO_2$ gas (>99.995%) by flowing the $CO_2$ through a stainless needle at a rate of 100 ml min$^{-1}$ over 5 min. This was followed by the addition of 0.75 ml of 100 mM sodium methanethiolate ($CH_3SNa$) with or without 1.86 ml of CO gas (1 atm, 99.9% purity). After rotating the bottle at 60 rpm for 7 days at room temperature (25 ± 2 °C), the headspace gas was analyzed by GC (Fig. 2b–d, Supplementary Figs. 19–23 and Supplementary Tables 3 and 4). The aqueous suspension was then centrifuged (10,000 rpm, 2 min) and measured for pH by a portable pH meter (Seven2Go Pro, Mettler Toledo).

We did not carry out "one-pot" organic synthesis in an electrochemical cell because $H_2$ evolution during sulfide electrolysis[15] prevented us from keeping methanethiol in the cell in the reaction period due to pressurization. As discussed in the main text, however, occurrence and combination of multiple reactions are likely to be more realistic than a single-pot reaction in natural hydrothermal vent environments.

**Competitive adsorption of CO with H₂S.** Sulfide sample (50 mg for each) was sealed in a serum bottle (19.5 ml) with a Teflon-laminated butyl rubber cap and an aluminum stopper in a globe box filled with $N_2$ and $H_2$ gases (volume ratio, 96:4). The bottle was then filled with pure He gas (>99.99995%) by flowing the He through a stainless needle at a rate of 100 ml min⁻¹ over 5 min. This was followed by the addition of 1 ml of 300 mM $Na_2S$ and the subsequent addition of 1 ml of 450 mM HCl. After rotating the bottle at 60 rpm for 1 day at room temperature (25 ± 2 °C), the headspace gas was analyzed by GC (Supplementary Figs. 26 and 27 and Supplementary Table 5). No carbon-sulfur compounds (for example, methanethiol, carbon disulfide, carbonyl sulfide) were detected.

From the pH values of aqueous suspensions (7.5–8.0; Supplementary Table 5), the initial dissolved $H_2S$ and $HS^-$ concentrations are predicted to be 90–120 mM in total, equilibrium $H_2S$ distribution being assumed in the gas and liquid phases without the surface adsorption.

**Sample analysis.** Inorganic gases, including $H_2$, CO, and $CH_4$, were quantified by a Shimadzu GC system equipped with a BID-2010 Plus detector (Tracera). A MICROPACKED-ST column (Shinwa) was attached. He (>99.99995%) was used as the carrier gas at a column flow rate of 7 ml min⁻¹. The column temperature was initially kept at 35 °C for 2.5 min, raised to 250 °C at a rate of 20 °C min⁻¹, and then raised to 265 °C at a rate of 4 °C min⁻¹. A chromatogram for a standard gas sample is shown in Supplementary Fig. 14 with obtained calibration curves.

Gas-phase sulfur compounds were analyzed by using a Shimadzu gas chromatograph-mass spectrometer (GCMS-QP2010 Ultra) equipped with a DB–SULFUR SCD column (60 m, 0.32 mm I.D, Agilent). He (>99.99995%) was used as the carrier gas at a column flow rate of 33.2 cm s⁻¹. The column temperature was initially kept at 35 °C for 5 min, raised to 155 °C at a rate of 10 °C min⁻¹, and then raised to 235 °C at a rate of 20 °C min⁻¹. The scan range was set to $m/z$ 5–200 Da. Compounds were identified by comparing the observed retention times and mass spectra with those of respective standards prepared from commercial reagents, and were quantified based on calibration curves (Supplementary Fig. 15). Several gas samples were also analyzed by the Tracera GC system with a DB–SULFUR SCD column at the measurement condition described in this paragraph (Supplementary Fig. 16).

Aqueous-phase products were characterized, after filtration with a polytetrafluoroethylene membrane filter (pore size, 0.2 μm), by using a Shimadzu HPLC system equipped with an electric conductivity detector and an anion exchange column (Shim-pack SCR-102H, Shimadzu) set at 40 °C. The $p$-toluenesulfonic acid aqueous solution (5 mM) was used as the eluent at a rate of 1.6 ml min⁻¹. A chromatogram for several organic acids is shown in Supplementary Fig. 17 with the obtained calibration curves.

Note that basified aqueous samples stored in non-airtight containers over several days occasionally showed 1–10 μM of acetate and pyruvate signals (data not shown) because basification facilitates absorption of organic acids from ambient air. To avoid crucial contaminations such as pyruvate, we set our detection cutoff at 0.01 mM and analyzed all aqueous samples as soon as they were prepared.

**Quantification of organosulfur products.** The four organosulfur compounds observed in the present study (methanethiol, MTA, dimethyl sulfide, and dimethyl disulfide) distribute both the gas- and aqueous-phases with significant fractions. Thus, aqueous-phase concentrations of methanethiol, dimethyl sulfide, and dimethyl disulfide were calculated from the partial pressures of respective compounds quantified by GC with the Henry's law constants reported in the literature (0.38, 0.56, and 0.58 mol l⁻¹ atm⁻¹, respectively)[50]. The constant for MTA was determined experimentally to be 5.8 ± 1.0 mol l⁻¹ atm⁻¹ from three independent measurements of the water-gas partitioning of MTA in a closed glass vial containing a certain volume of 100 mM phosphate buffer solution (pH 6.7). The constants for methanethiol, dimethyl sulfide, and dimethyl disulfide determined by this method were in agreement with the literature values with the percentage errors less than 15%.

The organosulfur compounds may also be present on the sulfide surfaces. Actually, when 1 M phosphoric acid ($H_3PO_4$) was added after the 7-day interaction of methanethiol with the NiS_PERM formed at –1.0 V_SHE, the gas-phase concentrations of MTA and methanethiol increased by 30 and 70%, respectively, while that of dimethyl sulfide decreased by 50%. Nevertheless, because such chemical processes may cause undesirable reactions that consume MTA, we did not take account of the surface adsorption in the yield determination of organosulfur compounds. For the same reason, in the nonenzymatic thioester synthesis experiment, we report the amounts of inorganic gases ($H_2$, CO, and $CH_4$) quantified from the gas-phase analysis alone (Supplementary Figs. 19b, 22, and 23, Supplementary Tables 3 and 4).

Dimethyl disulfide was always observed in the presence of methanethiol with the amount typically less than a few percent of the initial amount of methanethiol (Supplementary Figs. 19a and 20). A probable cause of the dimethyl disulfide formation is methanethiol oxidation by atmospheric $O_2$ (2 $CH_3SH$ + 0.5 $O_2$ → $CH_3SSCH_3$ + $H_2O$) at the timing of sample injections into the GC system owing to air intrusion (Supplementary Fig. 19b). Except for dimethyl disulfide, no oxidative organosulfur compound was detected (Supplementary Fig. 20). It was also confirmed that MTA, dimethyl sulfide, and dimethyl disulfide were stable in air.

**Thermodynamic calculation.** Redox potentials ($E_h$) of the $H^+/H_2$ and the $CO_2/$ CO couples as functions of temperature and pH (Supplementary Fig. 1) were calculated with the following equations:

$$E_h \frac{2}{2F}\left(2RT \ln a_{H^+} - RT \ln a_{H_2} - \Delta_f G^\circ\left(H_2\right)\right) \tag{4}$$

$$E_h = \frac{1}{2F}\left(\begin{array}{c}2RT \ln a_{H^+} + RT \ln a_{co_2} - RT \ln_{a_{co}} + \\ \Delta_f G^\circ\left(CO_2\right) - \Delta_f G^\circ\left(CO\right) - \Delta_f G^\circ\left(H_2O\right)\end{array}\right) \tag{5}$$

In these equations, $T$, $R$, and $F$ stand for temperature in kelvins, the gas constant (8.3145 J mol⁻¹ K⁻¹), and the Faraday constant (96,485 J mol⁻¹ V⁻¹), respectively. $\alpha_i$ represents the activity of the species $i$ that was calculated either using the extended Debye-Hückel equation[51] for aqueous ionic species ($H^+$) or setting the activity coefficient to unity for aqueous neutral species ($H_2$, $CO_2$, and CO). $\Delta_f G^\circ(i)$ signifies the standard Gibbs energy of formation of the species $i$ at the temperature and pressure of interest, which were calculated according to the revised HKF equations of state[52] using the relevant thermodynamic data and the revised HKF parameters reported in the literature[53,54]. $\Delta_f G^\circ$ of $H_2O$ was taken from Helgeson and Kirkham[55].

The $E_h$–pH diagrams of the sulfide–metal systems (Fig. 1b and Supplementary Fig. 4) were computed with the Act2 program in Geochemist's Workbench version 10.0.5 by using the thermodynamic dataset for aqueous species calculated by the above procedures, those for sulfide minerals listed in Supplementary Table 3 presented by Kitadai et al.[16], and those for pure metals compiled in Robie and Hemingway[56].

## Data availability
All data needed to evaluate the conclusions of this study are presented in this article, the Supplementary Information and/or the Supplementary Movie. Additional data related to this study are available from the corresponding author on reasonable request.

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

## Acknowledgements

We thank Reiko Nagano and Atsuko Fujishima for their help in laboratory experiments. This study was supported by JSPS KAKENHI (Grant Nos. 18H04456 and 20H00209) and the Astrobiology Center Program of NINS (Grant No. AB292004).

## Author contributions

N.K and Y.O. conceived the whole project based on the principled approach[57] and N.K. realized nonenzymatic thioester synthesis. Y.T. measured XANES spectra of the sulfide samples. S.O. performed SEM-EDX analysis, and W.T. prepared figures for the SEM-EDX data. N.K. performed all experiments except for the XANES and SEM-EDX measurements with technical support of Y.N., and prepared all figures and tables for the obtained results. S.O. considered possible intermediate steps in the MTA formation. N.K and Y.O. wrote the manuscript with input from R.N., M.Y., S.O., W.T., Y.N., Y.T., and K.T.

## Competing interests

The authors declare no competing interests.
