## [Peer Review File · Communications Chemistry]

Reviewers' comments:

Reviewer #1 (Remarks to the Author):

This is an important paper that should be published. The work is beautiful, elegant analytical chemistry, and partially answers a difficult question that goes back a long way in the origin of life field - at least to Christian de Duve and Hyman Hartman, who probably both deserve a namecheck in the references. In essence, thioesters, notably acetyl CoA, sit at the centre of cellular metabolism; they are the main end-product of 5 out of the 6 autotrophic pathways of CO₂ fixation, and also bridge between carbon and energy metabolism - phosphorylation of acetyl CoA gives acetyl phosphate, which can phosphorylate ADP to ATP. The simplest plausible prebiotic thioester is methyl thioacetate, which has to my knowledge never previously been synthesised, certainly not at the kinds of yields achieved here (there is a paper by Huber and Wachtershauser, cited by the authors, that claims to have synthesised 'activated acetate', meaning methyl thioacetate, but I believe they inferred methyl thioacetate as a likely intermediate on route to acetate). The issue is that thioesters are reactive (again as confirmed here) so for them to play a central role in protometabolism requires that they be synthesised faster than they are broken down. That has proved intractable to date, and that is why this paper is important: it shows that it can be done, and moreover in quite a simple way.

My reservations about this paper are more subtle and could be ignored, but I do think it would be helpful for the authors to address them. In essence, the authors impose an overpotential of between -0.5V and -1V. The synthesis of methyl thioacetate works well at or below -0.8V, but much less well at -0.5V. So the question is whether this voltage is realistic. The paper would benefit from a clearer exposition of the context here. There are also some missing citations that ought to be included. I'm thinking in particular of Roldan et al (Chem Commun 2015), who used an overpotential of -1V on nanocrystals of FeS (greigite) to form formate, acetate and pyruvate among other products. That was important work, but was criticised for having used an electrical voltage that is not directly equivalent to hydrothermal systems. The same criticism could be levelled here, and while the authors do address it, I felt they misconstrued the context on several occasions.

Is a -1V potential realistic? Some of these authors were part of an important earlier paper showing that hydrothermal vents can generate electricity. But my recollection of that paper is that the vent system they measured was a black smoker type system. In any case, the voltage measured was about 600 mV, from around -200 mV to about +400 mV, reflecting an oxygenated modern system. While 0.6V is important, -200 mV falls a long way short of driving the chemistry discussed in this paper. But that is not really made clear. It should be. So a second question is: how low a voltage could the more reducing alkaline hydrothermal vent systems sustain? Here the authors cite a Ken Nealson paper on the Cedars in northern California, but the Eh measurements in this terrestrial system were in the range -500 to -600 mV, which again are not sufficiently low to drive the chemistry reported here. The problem is addressed briefly in SI Fig 1 but that is basically theoretical. A major factor would be the concentration of H₂, which depends largely on its solubility and so pressure, so deep-marine hydrothermal systems such as Lost City ought to be more strongly reducing (with H₂ concentrations of ~15-20 mg/kg). To my knowledge accurate measurements of the Eh of fluids at Lost City have not been published, but Boyd et al (Phil Trans R Soc A, 2019) calculate about -0.8V for Lost city, based on measurements of pH, temperature and H₂ partial pressure. If correct, this value of -0.8V would indeed be able to drive the chemistry discussed here. It's also worth noting that the theoretical basis of the reduction potentials discussed here (pH dependence of redox potential) was recently proved experimentally for the first time (to my knowledge) by Hudson et al (PNAS, 2020), which again ought to be cited here (this paper appeared recently, after the current MS was submitted).

I reiterate that this context does not detract from the paper but would help to clarify the issues involved.

I have a number of other specific points, which I list briefly in chronological order:

Abstract - the value of 56% given for FeS and Co(Ni)S is correct for Co(Ni)S but not for FeS, where the proportion appears to be less than 30% from Fig 2C (which is where the 56% comes from). In any case, the term 'selectivity of 56%' was not clear to me in the Abstract and should be clarified. There was also no indication of prior yield here.

Line 53: "These enzymatic processes often remind us of their prebiotic origins"... seems a little too colloquial to me. The MS could do with editorial polishing in places.

Line 59 - thioesters: the citation to Goldford et al here is appropriate, though I would like to have seen citations to the much earlier work of Hartman (going back to the 1970s) and de Duve (going back to the late 1980s). de Duve especially laid out an important perspective that should not be forgotten.

Line 60 - entry point of phosphate into metabolism. This specific reaction has been reported by Whicher et al (OLEB 2016) who curiously found that thioacetate could readily be phosphorylated to acetyl phosphate in water under mild hydrothermal conditions (and that acetyl phosphate will in turn phosphorylate ADP to ATP under similar conditions) but that methyl thioacetate did not work as well. Nonetheless this paper is pertinent here.

Line 62-63 "The CO₂-to-CO reduction requires a highly reducing potential... that is inaccessible... in the absence of flavin-based electron bifurcation". This is an over-simplification. It can also be achieved by proton-motive membrane-proteins such as Ech, as elaborated in a number of papers, most recently and explicitly (with detailed redox calculations) by Vasiliadou et al (R Soc Interface focus, 2019). This hypothetical mechanism has also been validated experimentally by Hudson et al (PNAS, 2020) as noted above.

Line 69 Huber and Wachtershauser... I'm not sure where that figure came from, but I was under the impression that they had inferred the mechanism rather than demonstrated it. Perhaps there is some SI that I overlooked.

Line 75, electrically conducting vents. If I'm not mistaken these measurements were made on black-smoker type systems, in any case as noted above, between oxygenated surroundings and moderately reducing hydrothermal fluids, with a potential difference of about 600 mV (-200 mV to + 400 mV). While a proof of concept (FeS minerals are semiconducting) these conditions are far more oxidising than the conditions discussed in the paper. For the ancient oceans with CO₂ as the main oxidant the potential difference would be much more limited.

Line 104 - the text is initially a bit cagey about exactly what the electrical potential was, although this becomes clear later on. Given that the Methods are at the end it would be helpful to specify the range of potentials up front.

Line 118. I'm not sufficiently familiar with XANES as a technique but it was not obvious to me how the 12% was inferred here. More details on how the 12% value was calculated would be helpful. The requisite information appears to be in Fig S11 but the quality of fitting there seems a little equivocal to me (based on the baseline subtraction). I'm sure this reflects my ignorance of the methodology but it would be helpful for others too to provide more detail as it is a critical point in the paper.

Likewise in Fig 1a (XRD patterns) it is not clear to me that there is any peak corresponding to NiO. Line 123 seems to confirm that there isn't a peak, implying that NiO is finely dispersed. That may be so, but again this places the onus on proving that the 12% figure is accurate, which again requires a clearer explanation of the XANES derivations.

Fig 1d - the S peak does not change at all, so I assume this is kept constant and the other peaks are normalised to that. But this is not stated anywhere. If normalised this should be made clear (and if not, why does the S peak remain exactly the same, while Ni shifts?)

Line 145 "Cocoprecipitation with FeS led to decline of the amounts of CO on NiS and CoS (Figure 2a) probably because of the decrease in the surface reactive sites." This makes sense but again has potentially important prebiotic ramifications. The freshly precipitated metal sulfides were 50:50 mixtures of Ni:Fe but my understanding is that Fe is likely to have been ~10-fold more abundant in Hadean oceans. If this more realistic ratio were reflected in the precipitates subjected to electrolysis it is doubtful that any CO would have been detected at all. It is important to acknowledge that this study proves the principle that redox differences across vent walls are capable of reducing metal sulfides to native metals, which are capable of reducing CO₂ to CO, etc - it is demonstrated beautifully in this system, but the actual voltage, the actual precipitate compositions and (later) the actual concentrations of CO and CH₃SH are all generous and may well be at the extreme ends of those found in nature. The paper would be better if it were clear about this.

line 158 - I was wondering if there was any detection of thioacetate here.

Figure 2a - the CO concentration with Fe(Ni)S is negligible at even -0.8V here, despite the Fe:Ni ratio being 1:1. Even with -1V, little CO forms with this most realistic of compositions (reflecting the Hadean ocean). This is why I suspect that if a 10:1 ratio of Fe:Ni had been used then CO concentration would have been undetectable. So I would say this is a proof of principle and does not reflect realistic Hadean conditions.

Fig 2b - again NiS works well but Fe(Ni)S much less well, despite a modest excess of methanethiol per sulfide. With Fe(Ni)S, yield is close to zero at -0.8V, the generous Eh for Lost City today. All this ought to be acknowledged.

Fig 2c - here it seems as if Fe(Ni)S was successful as the per cent conversion of CO to methyl thioacetate (MTA) is around 20-30% even at lower potentials (-0.7V) and this is reported (erroneously) in the abstract. But in fact it only appears to be good because there was so little CO reduced on this surface in the first place. This may give a helpful indication of the 2-step reaction mechanism but the straight claim seems misleading to me and ought to be modified.

Fig 2d - with added CO. Here the pattern for Fe(Ni)S is surprising and implies that it is adsorbing onto other metal ions than simply native Ni (which reduces CO₂ to CO). This was not commented on much in the paper. Apart from anything else it implies that CO formed elsewhere may well be trapped within the system by adsorbing onto more surfaces.

Figure legend - again one is entitled to wonder where all this methanethiol is coming from. It has been detected in hydrothermal systems but the consensus, insofar as there is one at all, is that it is derived from thermal decomposition of organic molecules buried deeper in the crust. So not available at the origin of life. This is contentious I know; but again the issue would be solved by presenting this work as proof of principle rather than 'solving' the problem. I would say it is closer to showing how the problem could be solved.

Line 172 - I am not sure what the multiple numbers are referring to here, even when consulting Fig S25. I assume w/o externally added CO/MTA. If so, it is surprising to me that much more acetate than formate appears to be synthesized in the absence of both added CO and CH₃SH. That seems improbable so I suspect I am misunderstanding something. At least this needs to be made clear.

Line 187/88 - steep decline in yield of MTA below -0.8V - which is borderline the values likely to be

achieved in vents. It might be worth mentioning that the existence of awaruite in many fossil vent systems implies very high partial pressure of H₂, which would certainly lower the reduction potential to the required range (see e.g. Vasiliadou et al).

Line 210 "(Fe,Ni)S produced a considerable amount of MTA from initially introduced CO and methanethiol even after the -0.6 VSHE electrolysis (Figure 2d)." As noted above this is only because the amount of CO generated at -0.6V was negligible. So a modest proportion of nearly nothing.

Line 226 Potential level for NiO formation... depends a lot on how much Ni²⁺ was available relative to Fe²⁺; I think much less than implied here.

As an aside here it would also be good to clarify that 'less than or equal to -0.6V would mean -0.7V, not -0.5V. This is strictly correct as written but has potential to mislead, as 'less than' might imply a less reducing (i.e. more positive) reduction potential. It would be useful to explicitly state that the use of less than or equal to refers to a more extreme reduction potential, more strongly negative, less achievable, whereas in the context it implies that the conditions could be even more mild than stated.

Line 236: "Thus, the thioester synthesis via the formation of Ni sulfide PERMs should have been robustly concomitant with ubiquitous hydrothermal activities on the primordial seafloor." To my mind this statement is too strong, given my comments above. I think the study proves the principle and the conditions required are quite mild and potentially overlap with those on the primordial seafloor. But the phrase at present is misleading: to form thioesters under these conditions requires very low reduction potentials (lower than -0.8V, which is the lowest that has been detected in modern systems) coupled with high relative Ni²⁺ concentrations (or Co maybe), coupled with a high stoichiometric flux of methanethiol (which is highly implausible). So I think we would be deluding ourselves if we thought that this paper solves the whole problem, but it is important nonetheless because it shows what is possible. The conditions are right on the cusp of being realisable, and call for more work.

Line 251 - Deep sea hydrothermal systems are referred to Fig S1 but there is nothing in that figure on pressure. Pressure would of course increase the partial pressure of H₂, lowering the reduction potential, making the conditions required more realistic. Again, this has not been clearly stated in the paper.

Fig S1 - I assume these calculations are based on the standard hydrogen electrode with 1 atmosphere pressure of H₂ (hence about 0.74 mg/kg dissolved H₂, well below those at Lost City. This could do with expanding.

I won't comment more on the SI figures, except to call again for a little more detail on the XANES derivatizations. I would like to say though that these are really beautiful data, a joy to behold how well this work has been done. The team are to be congratulated on the quality of their data.

That's all. I hope my comments are taken in the spirit they are intended, to improve (a little) an excellent paper that clearly deserves publication in Commun Chem, and should excite a wide readership in answering a long-standing, difficult and important question. It ought to be highly cited.

I am happy for the authors to know that I am
Nick Lane

Reviewer #2 (Remarks to the Author):

The manuscript entitled 'Thioester synthesis through geoelectrochemical CO₂ fixation on Ni sulfides' by Kitadai et al. presents results about the catalytic properties of the NiS/Ni(0) system to reduce CO₂ to CO and finally the reaction with thiols to corresponding thioesters, here S-methyl thioacetate. This is an interesting process, which might be a puzzle piece in the emergence of life or at least an early step to the acetyl-CoA pathway, which is the energy-releasing route of biological CO₂ fixation.

This adds to already known and well described properties of Fe and Ni sulfides (Wachtershauser and co-authors), which were extensively studied in the past as catalysts. The new findings are interesting in particular in the context of the observed spontaneous generation of electricity in deep-sea vent chimneys and mineral deposits (Yamamoto et al.).

Another important aspect is that there is still a gap of knowledge in the explanation of the thioester-dependent acetyl-CoA metabolism.

In general, there are many and detailed aspects in the context of the emergence of life, which are of interest for the broader community. The experiments might trigger further investigations on reactions, where these sub stoichiometric sulfides may play a role as catalysts.

Here are my comments:

A critical point is the aging of the NiS samples in the partially reduced state. The authors should comment on the long-term stability of the synthesized particles and the change of the catalytic properties.

For the curiosity, is there a size-dependency observed in the XANES measurements?

I expect that there are phase transitions in the NiS upon reduction. Is there any evidence for phase transition, which could cause the formation of highly active sites?

In the experimental section of the supplementary information there is a comment about the glovebox that 4% of hydrogen gas is used to avoid oxidation. It is important to prove that the prepared particles do not have absorbed hydrogen. I understand that the experimental prove and potential desorption might be difficult, but I would recommend the use of D₂O in the CO conversion to S-methylthioacetate.

Please explain in more detail the mentioned CO-to-MTA reaction efficiencies because this is only a qualitative measure. If possible, give numbers, in which way the 'efficiency' was improved. Same holds for the 'considerable amount of MTA' in line 210.

In lines 211 and 212 there is reference given to Figures 7 and 8. Please add in the supplementary information.

A minor point, which requires some better definition and chemical description is the term FeS_{Perm} for the FeS, partially electro reduced to metal. I think this could be more precisely expressed in a formula e.g. Fe_xS, defining $x > 1$ or Fe_{>1}S, where x and >1 are subscript.

Reviewer #3 (Remarks to the Author):

It is a beautiful work and an interesting hypothesis as an effort to understand the electrochemical reactions in the early earth. The experimental design is well-established and this work can provide many insights for the future studies. For the publication, the following points should be improved and addressed.

1. The results are fascinating and contains a lot of new progress but the logical flow in the

manuscript is a little bit confusing. In fig 1 and fig 2, the electrochemical analysis is done for all metal sulfides. We understand that it is a kind of the screening to identify the excellent activity of NiS. But for the readers, the flow seems not focused. As the author wanted to emphasize the Ni, it would be better to move some data to supporting information.

2. In this regards, the reason why Ni containing sulfide is better than Co, Fe sulfide is not clear. Also, it was mentioned that NiS with Co and Fe is the best. But the scientific discussion should be added in the details.

3. All the Ni on the surface of NiS during the electrolysis can be bound with CO? Quantitative analysis can be necessary to get the number of active site of Ni for the CO attachment.

4. If the methanethiol exist together during the CO₂-CO electrochemical conversion, the yield to thiolester can be decreased a lot? Why is the separate step necessary?

5. In produc analysis, some organic molecules such as pyruvate were detected. But the mechanism is not clear.

6. In the mantle, there is also FeS. The competeition between Fe and Ni for the CO₂/CO binding will affect or we can consider the cooperative mechanism.

7. Is the concentration gradient of sulfur and hydroggen enough to drive the electrochemical reduction of CO₂ to C?

8. Ni or transition metals can act as a catalyst for the reaction of CO and methanethiol. Is it unexpected?

Response to Reviews

Responses to the Reivewer 1's comments

1-1. This is an important paper that should be published. The work is beautiful, elegant analytical chemistry, and partially answers a difficult question that goes back a long way in the origin of life field – at least to Christian de Duve and Hyman Hartman, who probably both deserve a namecheck in the references. In essence, thioesters, notably acetyl CoA, sit at the center of cellular metabolism; they are the main end-product of 5 out of the 6 autotrophic pathways of CO₂ fixation, and also bridge between carbon and energy metabolism – phosphorylation of acetyl CoA gives acetyl phosphate, which can phosphorylate ADP to ATP. The simplest plausible prebiotic thioester is methyl thioacetate, which has to my knowledge never previously been synthesized, certainly not at the kinds of yields achieved here (there is a paper by Huber and Wächtershäuser, cited by the authors, that claims to have synthesized ‘activated acetate’, meaning methyl thioacetate, but I believe they inferred methyl thioacetate as a likely intermediate on route to acetate). The issue is that thioesters are reactive (again as confirmed here) so for them to play a central role in protometabolism requires that they be synthesized faster than they are broken down. That has proved intractable to date, and that is why this paper is important: it shows that it can be done, and moreover in quite a simple way. My reservations about this paper are more subtle and could be ignored, but I do think it would be helpful for the authors to address them.

Response

We are grateful to Reviewer 1 (Professor Nick Lane) for his encouraging report on our manuscript. As detailed below (from No. 1-1 to 1-29), we have revised our manuscript following his very useful criticisms and suggestions.

Christian de Duve and Hyman Hartman are indeed the key scientists in the topic examined here. They must have been mentioned in our manuscript. In contrast, the phylogenetic conservation of CODH/ACS, which was mentioned in our original Introduction, remains controversial (Inoue et al., 2019; Berkemer and McGlynn, 2020). Thus, we removed the related description and cited the works by Duve and Hartman in the revised Introduction as follows:

“Thioesters occupy the central position in cellular metabolism as exemplified by acetyl-CoA. This fact has suggested thioesters to be the crucial components of prebiotic precursor of metabolism since as early as 1960’s (for example, Hartman, 1975; de Duve, 1991; see also

Response to Reviews

Wächtershäuser, 1992). A key enzyme in thioester biosynthesis is carbon monoxide dehydrogenase/acetyl-CoA synthase (CODH/ACS), in which the Ni-based active centers reduce CO₂ to CO, react CO with a methyl group to form acetyl, and acetylate CoA to form acetyl-CoA. This apparently simple process with pivotal Ni has evoked the idea that a prebiotic CO₂ fixation on a (Fe,Ni)S mineral preceded the CODH/ACS-based reaction (Russell and Martin, 2004; Sousa et al., 2013).”

(References)

- Berkemer, S. J. & McGlynn, S. E. A new analysis of Archea–Bacteria domain separation: Variable phylogenetic distance and the tempo of early evolution. *Mol. Biol. Evol.* **37**, 2332–2340 (2020).
- de Duve, C. *Blueprint For a Cell: The Nature and Origin of Life* (Neil Patterson Publishers, 1991).
- Hartman, H. Speculations on the origin and evolution of metabolism. *J. Mol. Evol.* **4**, 359–370 (1975).
- Inoue, M. et al. Structural and phylogenetic diversity of anaerobic carbon-monoxide dehydrogenase. *Front. Microbiol.* **9**, 3353 (2019).
- Russell, M. J. & Martin, W. The rocky roots of the acetyl-CoA pathway, *TRENDS Biochem. Sci.* **29**, 358–363 (2004).
- Sousa, F. L. et al. Early bioenergetic evolution. *Phil. Trans. R. Soc. B.* **368**, 20130088 (2013).
- Wächtershäuser, G. Groundworks for an evolutionary biochemistry: The iron–sulphur world. *Prog. Biophys. Molec. Biol.* **58**, 85–201 (1992).

1-2. In essence, the authors impose an overpotential of between -0.5 V and -1 V. The synthesis of methyl thioacetate works well at or below -0.8 V, but much less well at -0.5 V. So the question is whether this voltage is realistic. The paper would benefit from a clearer exposition of the context here. There are also some missing citations that ought to be included. I’m thinking in particular of Roldan et al. (Chem Commun 2015), who used an overpotential of -1 V on nanocrystals of FeS (greigite) to form formate, acetate and pyruvate among other products. That was important work, but was criticized for having used an electrical voltage that is not directly equivalent to hydrothermal systems.

Response to Reviews

The same criticism could be levelled here, and while the authors do address it, I felt they misconstrued the context on several occasions.

Response

We added a discussion on the geoelectric potentials available on the primitive seafloor, following the reviewer's suggestions given at No. 1-3, 1-22, and 1-27 (page 11 of Highlighted Revision):

“The electric potential required for these reactions ($\leq -0.6 V_{SHE}$; Figures 1 and 2) is more reducing than that observed in a present-day black smoker system ($\geq -0.022 V_{SHE}$; Yamamoto et al., 2017). However, the value of $-0.6 V_{SHE}$ and even lower potentials are available in H_2 -rich alkaline hydrothermal fields (Figure S1) (Morrill et al., 2013; Boyd et al., 2020). Deep-sea systems are more advantageous than terrestrial ones because the elevated pressure increases the solubility of H_2 , which indeed serves as an electron source for the reduction of seawater CO_2 in the presence of (Fe,Ni)S precipitates (Hudson et al., 2020). On the primordial ocean floor, widespread occurrence of elevated hydrothermal H_2 fluxes has been suggested from the presence of awaruite in many fossilized serpentinization systems (Rajendran and Nasir, 2014; McCollom, 2016).”

Roldan et al. (2015) examined CO_2 electroreduction on greigite (Fe_3S_4) in an H-type cell that had two compartments separated by a dialysis tubing (TUB2014; the molecular weight cut-off (MWCO) = 12,000 to 14,000 Daltons), with a carbon working electrode placed at one side, and a Ag/AgCl reference electrode and a platinum counter electrode at the other side (see their Figure S1). The separate placement of the working and reference electrodes, and the close placement of the reference and counter electrodes, should significantly influence the potential control and measurement during the CO_2 electrolysis. In addition, dissolved species should readily diffuse into both the working and counter electrode sides due to the large MWCO of dialysis tubing. Thus, the reported reaction products may not reflect the catalytic capability of Fe_3S_4 under the condition described in the method section. In addition, the supplementary NMR data (Figure S18–S20) shows many contaminant signals with the intensities even larger than the identified compounds. Because there is no figure BEFORE the reactions, we are not very sure that the reported products are in reality contamination or not. To our best knowledge, we have not seen any follow-up experiments of theirs. Actually, we observed

Response to Reviews

no organic compound formation in our experiment following their methods. Because of these reasons, we would like to refrain from commenting on Roldan et al. (2015).

(References)

Hudson R. et al. CO₂ reduction driven by a pH gradient. *Proc. Natl. Acad. Sci. U.S.A.* **117**, 22873–22879 (2020).

McCollom, T. M. Abiotic methane formation during experimental serpentinization of olivine. *Proc. Natl. Acad. Sci. USA* **113**, 13956–13970 (2016).

Morrill, P. L. et al. Geochemistry and geobiology of a present-day serpentinization site in California: The Cedars. *Geochim. Cosmochim. Acta* **109**, 222–240 (2013).

Rajendran, S. & Nasir, S. Hydrothermal altered serpentinized zone and a study of Ni-magnesioferrite-magnetite-awaruite occurrences in Wadi Hibi, Northern Oman Mountain: Discrimination through ASTER mapping. *Ore Geology Rev.* **62**, 211–226 (2014).

Roldan, A. et al. Bio-inspired CO₂ conversion by iron sulfide catalysts under sustainable conditions. *Chem. Commun.* **51**, 7501–7504 (2015).

Vasiliadou, R., Dimov, N., Szita, N., Jordan, S. F. & Lane, N. Possible mechanisms of CO₂ reduction by H₂ via prebiotic ventral electrochemistry. *Interface Focus* **9**, 201290073 (2019).

Yamamoto, M. et al. Spontaneous and widespread electricity generation in natural deep-sea hydrothermal fields. *Angew. Chem. Int. Ed.* **56**, 5725–5728 (2017).

1-3. Is a –1 V potential realistic? Some of these authors were part of an important earlier paper showing that hydrothermal vents can generate electricity. But my recollection of that paper is that the vent system they measured was a black smoker type system. In any case, the voltage measured was about 600 mV, from around –200 mV to about +400 mV, reflecting an oxygenated modern system. While 0.6 V is important, –200 mV falls a long way short of driving the chemistry discussed in this paper. But that is not really made clear. It should be. So a second question is: how low a voltage could the more reducing alkaline hydrothermal vent systems sustain? Here the authors cite a Ken Nealson paper on the Cedars in northern California, but the Eh measurements in this terrestrial system were in the range –500 to –600 mV, which again are not sufficiently low to drive the chemistry reported here. The problem is addressed briefly in SI Fig 1 but that is basically theoretical. A major factor

Response to Reviews

would be the concentration of H₂, which depends largely on its solubility and so pressure, so deep – marine hydrothermal systems such as Lost City ought to be more strongly reducing (with H₂ concentrations of ~15–20 mg/kg). To my knowledge accurate measurements of the Eh of fluids at Lost City have not been published, but Boyd et al. (Phil Trans R Soc A, 2019) calculate about –0.8 V for Lost City, based on measurements of pH, temperature and H₂ partial pressure. If correct, this value of –0.8 V would indeed be able to drive the chemistry discussed here. It's also worth noting that the theoretical basis of the reduction potentials discussed here (pH dependence of redox potential) was recently proved experimentally for the first time (to my knowledge) by Hudson et al. (PNAS, 2020), which again ought to be cited here (this paper appeared recently, after the current MS was submitted). I reiterate that this context does not detract from the paper but would help to clarify the issues involved.

Response

We appreciate the helpful comments and references given by this reviewer here. In the discussion on the geoelectric potential (see our response No. 1-2), we specified that the potential level observed by Yamamoto et al. (2017) in a present-day black smoker system is insufficient to drive the reactions demonstrated in this study, and cited the works by Boyd et al. (2019) and Hudson et al. (2020) as key pieces of evidence supporting the existence of geoelectrochemical activities sufficient for the abiotic thioacetate synthesis on the primordial seafloor.

Our thermodynamic calculation (Figure S1) is indeed basically theoretical, not simulating any specific hydrothermal vent environment. But the calculated temperature (0–300°C) and pH (2–12) conditions and H₂ concentration (1 mmol kg⁻¹) are all within a range realistic in deep-sea hydrothermal systems. Furthermore, as pointed out by this reviewer, a major factor would be the concentration of H₂, which depends on pressure: so the high pressure in deep-sea simply shifts our theoretical results to a more favorable reducing direction. Thus, we keep our thermodynamic reasoning as well.

1-4. I have a number of other specific points, which I list briefly in chronological order:

Abstract – the value of 56% given for FeS and Co(Ni)S is correct for Co(Ni)S but not for FeS, where the proportion appears to be less than 30% from Fig 2C (which is where the 56% comes from). In

Response to Reviews

any case, the term ‘selectivity of 56%’ was not clear to me in the Abstract and should be clarified. There was also no indication of prior yield here.

RESPONSE

For clarity, we replaced the term ‘selectivity’ with ‘yield based on CO’, and specified the percentages achieved by (Fe,Ni)S and NiS as well as by (Co,Ni)S. In addition, we modified the first and last sentences of Abstract according to the revision made in the first paragraph of Introduction (see our response No 1-1), as follows:

“A prevailing scenario of the origin of life postulates thioesters as key intermediates in protometabolism, but there is no experimental support for the prebiotic CO₂ fixation routes to thioesters. Here we demonstrate that, under a simulated geoelectrochemical condition in primordial ocean hydrothermal systems (−0.6 to −1.0 V versus the standard hydrogen electrode), nickel sulfide (NiS) gradually reduces to Ni⁰, while accumulating surface-bound carbon monoxide (CO) due to CO₂ electroreduction. The resultant partially reduced NiS realizes thioester (S-methyl thioacetate) formation from CO and methanethiol even at room temperature and neutral pH with the yield up to 35% based on CO. This thioester formation is not inhibited, or even improved, by 50:50 coprecipitation of NiS with FeS or CoS (the maximum yields; 27 or 56%, respectively). Such a simple thioester synthesis likely occurred in Hadean deep-sea vent environments, setting a stage for the autotrophic origin of life.”

1-5. Line 53: “These enzymatic processes often remind us of their prebiotic origins” seems a little too colloquial to me. The MS could do with editorial polishing in places.

RESPONSE

This sentence was deleted in association with the revision of the first paragraph. See our response No 1-1.

1-6. Line 59 – thioesters: the citation to Goldford et al here is appropriate, though I would like to have seen citations to the much earlier work of Hartman (going back to the 1970s) and de Dube (going

Response to Reviews

back to the late 1980s). de Duve especially laid out an important perspective that should not be forgotten.

RESPONSE

We cited the works by Hartman and de Duve (Hartman, 1975; de Duve, 1991) in the sentence commented here:

“Thioesters ~~possibly played rather~~ **might have played** central roles in protometabolism not only as carbon sources but also as energy currencies in a manner analogous to ATP (Hartman, 1975; de Duve, 1991; Goldford et al., 2017), serving as an entry point of phosphate into metabolism (Sousa et al., 2013; Whicher et al., 2018).

(References)

de Duve, C. *Blueprint For a Cell: The Nature and Origin of Life* (Neil Patterson Publishers, 1991).

Goldford, J. E., Hartman, H., Smith, T. F. & Segre, D. Remnants of an ancient metabolism without phosphate. *Cell* **168**, 1126–1134 (2017).

Hartman, H. Speculations on the origin and evolution of metabolism. *J. Mol. Evol.* **4**, 359–370 (1975).

Sousa, F. L. et al. Early bioenergetic evolution. *Phil. Trans. R. Soc. B.* **368**, 20130088 (2013).

1-7. Line 60 – entry point of phosphate into metabolism. This specific reaction has been reported by Whicher et al (OLEB 2016) who curiously found that thioacetate could readily be phosphorylated to acetyl phosphate in water under mild hydrothermal conditions (and that acetyl phosphate will in turn phosphorylate ADP to ATP under similar conditions) but that methyl thioacetate did not work as well. Nonetheless this paper is pertinent here.

RESPONSE

We cited Whicher et al. (2016) in the sentence commented here. See our response No. 1-6.

Response to Reviews

(Reference)

Whicher, A., Camprubi, E., Pinna, S., Herschy, B. & Lane, N. Acetyl phosphate as a primordial energy currency at the origin of life. *Orig. Life Evol. Biosph.* **48**, 159–179 (2018).

1-8. Line 62–63 “The CO₂-to-CO reduction requires a highly reducing potential...that is inaccessible...in the absence of flavin-based electron bifurcation”. This is an oversimplification. It can also be achieved by proton-motive membrane-proteins such as Ech, as elaborated in a number of papers, most recently and explicitly (with detailed redox calculations) by Vasiliadou et al. (R Soc Interface Focus, 2019). This hypothetical mechanism has also been validated experimentally by Hudson et al. (PNAS, 2020) as noted above.

RESPONSE

This sentence was modified as follows.

“The CO₂-to-CO conversion is endergonic even with H₂ as the reductant (Figure S1), so biological CO production uses either electron bifurcation or a chemiosmotic pH gradient across the cell membrane to overcome the energy shortage.”

1-9. Line 69 Huber and Wächtershäuser... I’m not sure where that figure come from, but I was under the impression that they had inferred the mechanism rather than demonstrated it. Perhaps there is some SI that I overlooked.

RESPONSE

Huber and Wächtershäuser have described the MTA formation. See the right column on page 246 of their report.

(References)

Huber, C. & Wächtershäuser, G. Activated acetic acid by carbon fixation on (Fe,Ni)S under primordial conditions. *Science* **276**, 245–247 (1997).

Response to Reviews

1-10. Line 75, electrically conducting vents. If I'm not mistaken these measurements were made on black-smoker type systems, in any case as noted above, between oxygenated surroundings and moderately reducing hydrothermal fluids, with a potential difference of about 600 mV (–200 mV to +400 mV). While a proof of concept (FeS minerals are semiconducting) there conditions are far more oxidizing than the conditions discussed in the paper. For the ancient oceans with CO₂ as the main oxidant the potential difference would be much more limited.

RESPONSE

The natural sulfide chimney rocks conduct the electric potentials of internal hydrothermal fluids to the outer chimney wall with little potential loss (see Fig. 2a of Yamamoto et al., 2017) owing to the low electrical resistivity (0.11 to 4.97 Ω cm; Ang et al., 2015). Thus, as illustrated in Figure S1, geoelectricity generation through coupling between the H₂-to-H⁺ oxidation in alkaline fluids with CO₂-to-CO reduction and/or the metal sulfide_PEM formation at the vent-seawater interface is expected to be possible in primordial ocean hydrothermal systems. This theoretical basis was recently proved by Hudson et al. (2020), as informed by this reviewer at the No.1-3 comment. This work has been cited in the revised manuscript (see our response No. 1-2).

(References)

Ang, R. et al. (2015) Thermoelectricity generation and electron-magnon scattering in a natural chalcopyrite mineral from a deep-sea hydrothermal vent. *Angew. Chem. Int. Ed.* **54**, 12909–12913 (2015).

Hudson R. et al. CO₂ reduction driven by a pH gradient. *Proc. Natl. Acad. Sci. U.S.A.* **117**, 22873–22879 (2020).

Yamamoto, M. et al. Spontaneous and widespread electricity generation in natural deep-sea hydrothermal fields. *Angew. Chem. Int. Ed.* **56**, 5725–5728 (2017).

1-11. Line 104 – the text is initially a bit cagey about exactly what the electrical potential was, although this becomes clear later on. Given that the Methods are at the end it would be helpful to specify the range of potentials up front.

Response to Reviews

RESPONSE

We specified the potential range examined in this study as follows:

“The obtained sulfides were exposed to a constant electric potential (-0.5 to -1.0 V_{SHE} ; see Discussion for its realizability) for 7 days in 100 mM NaCl at room temperature ($25 \pm 2^\circ\text{C}$) ~~under continuous CO_2 bubbling that maintained the solution pH at 6 ± 0.25~~ (Figures S2 and S3). **The solution pH was maintained at 6 ± 0.25 by continuous CO_2 bubbling.**”

1-12. Line 108. I’m not sufficiently familiar with XANES as a technique but it was not obvious to me how the 12% was inferred here. More details on how the 12% value was calculated would be helpful. The requisite information appears to be in Fig S11 but the quality of fitting there seems a little equivocal to me (based on the baseline subtraction). I’m sure this reflects my ignorance of the methodology but it would be helpful for others too to provide more detail as it is a critical point in the paper.

RESPONSE

We added the following explanation of the XANES data analysis in Materials and Methods (page 4 of Supplementary Information):

“The percentages of zerovalent metal in the electrolyzed NiS and CoS were estimated by a least-squares fitting of the sample spectra after background-subtraction and normalization with an X-ray absorption spectroscopy data processing software ATHENA (Ravel and Newville, 2005). For the NiS samples, the linear combination of pure Ni^0 and the NiS electrolyzed at -0.5 V_{SHE} were used for the fitting between 8300 and 8440 eV. The NiS electrolyzed at -0.5 V_{SHE} was used as a reference of Ni_3S_2 based on the thermodynamic calculation (Figure 1b), XRD pattern (Figure 1e), and the XANES spectral profile (Figure S11a) identical to that of pure Ni_3S_2 (Van Loon et al., 2015). The energy region higher than 8440 eV was not analyzed because of lower signal-to-noise ratio and of greater influence from the local environment of Ni atom. The best fit was determined by calculating the lowest R factor, which was defined as:

$$R = \sum \{\mu_{exp}(E) - \mu_{cal}(E)\}^2$$

Response to Reviews

In this equation, μ_{exp} and μ_{cal} are the experimental and the calculated absorbances at a given energy E , respectively. For the CoS samples, the energy range 7690–7830 eV were fitted with the linear combination of pure Co^0 and the CoS electrolyzed at $-0.5 V_{\text{SHE}}$ in a manner similar as Ni described above.”

(References)

Ravel, B. & Newville, M. (2005) ATHENA, ARTEMIS, HEPHAESTUS: data analysis for X-ray absorption spectroscopy using IFEFFIT. *J. Synchrotron Rad.* **12**, 537–541 (2005).

Van Loon, L. L., Throssell, C. & Dutton, M. D. Comparison of nickel speciation in workplace aerosol samples using sequential extraction analysis and X-ray absorption near-edge structure spectroscopy. *Environ. Sci. Processes Impacts* **17**, 922–931 (2015).

1-13. Likewise in Fig 1a (XRD patterns) it is not clear to me that there is any peak corresponding to Ni^0 . Line 123 seems to confirm that there isn't a peak, implying that Ni^0 is finely dispersed. That may be so, but again this places the onus on proving that the 12% figure is accurate, which again requires a clearer explanation of the XANES derivations.

RESPONSE

In Figure S10, we have compared the XRD pattern of the NiS_PERM prepared at $-1.0 V_{\text{SHE}}$ and that of a mixture of pure Ni^0 and the NiS electrolyzed at $-0.5 V_{\text{SHE}}$ corresponding to heazlewoodite ($\text{Ni}^0 : \text{NiS}_{2/3} = 12 : 88$). No signal appeared at around 52° in the NiS_PERM's XRD pattern, indicating that Ni^0 did not form a localized crystalline domain in the Ni_3S_2 structure.

The XANES data analysis method has been added in the revised Materials and Methods. See our response No. 1-12.

Response to Reviews

1-14. Fig 1d – the S peak does not change at all, so I assume this is kept constant and the other peaks are normalized to that. But this is not stated anywhere. If normalized this should be made clear (and if not, why does the S peak remain exactly the same, while Ni shifts?)

RESPONSE

The explanation of peak normalization in Figure 1 caption was modified as follows:

~~“In (d), the S signal intensities (2.3 keV) for NiS before (blue) and after the electrolysis at $-0.5 V_{SHE}$ (green) or $-1.0 V_{SHE}$ (red) are matched with each other for comparison. In (d), the signal intensities for NiS before (blue) and after the electrolysis at $-0.5 V_{SHE}$ (green) or $-1.0 V_{SHE}$ (red) are normalized so that the S signal intensities (2.3 keV) become identical.”~~

For more clarity, we added a brief explanation in the revised manuscript (page 5 of Highlighted Revision):

“Further reduction of Ni_3S_2 at lower electric potentials was indicated by energy dispersive X-ray spectroscopy (EDS) mapping on the NiS particles electrolyzed at $-1.0 V_{SHE}$ (Figure 1c), where a clear decrease in the sulfur signal intensity relative to nickel was observed in comparison with pure NiS, and with the NiS electrolyzed at $-0.5 V_{SHE}$ (Figure 1d, in which all EDS data are scaled to have the same S signal intensities at 2.3 keV).”

1-15. Line 145 “Coprecipitation with FeS led to decline of the amounts of CO on NiS and CoS (Figure 2a) probably because of the decrease in the surface reactive sites.” This makes sense but again has potentially important prebiotic ramifications. The freshly precipitated metal sulfides were 50:50 mixtures of Ni:Fe but my understanding is that Fe is likely to have been ~10-fold more abundant in Hadean oceans. If this more realistic ratio were reflected in the precipitates subjected to electrolysis it is doubtful that any CO would have been detected at all. It is important to acknowledge that this study proves the principle that redox differences across vent walls are capable of reducing metal sulfides to native metals, which are capable of reducing CO_2 to CO, etc – it is demonstrated beautifully in this system, but the actual voltage, the actual precipitate compositions and (later) the actual concentrations of CO and CH_3SH are all generous and may well be at the extreme ends of those found in nature. The paper would be better if it were clear about this.

Response to Reviews

RESPONSE

The Ni/Fe ratio in sulfide deposits in primordial deep-sea vent environments may not have directly reflected the oceanic metal ion composition. In the revised manuscript, we discussed the availability of Ni-sulfides based on a reported field observation (Bekker et al., 2009) and on a thermodynamic calculation (page 11 of Highlighted Revision):

“In the Archean to Hadean eons, submarine hydrothermal environments may also have favored NiS precipitation owing to the huge supply of mantle-derived Ni into the ocean. The resultant massive Ni sulfide deposits are seen today in association with Archean komatiite with the Ni/Fe weight ratios occasionally exceeding one (Bekker et al., 2009). Given the estimated Ni²⁺ and Fe²⁺ concentrations in the early Archean seawater with their sulfide solubilities, selective NiS precipitation relative to FeS is expected in micro- to semimicro-molal level presence of hydrogen sulfide (H₂S and HS⁻) (Figure S26), which is a likely concentration range at the outer surface of ancient alkaline hydrothermal chimneys (Shibuya et al., 2016).”

Figure S26. Gibbs energies of reaction for the FeS (mackinawite) and NiS (millerite) formations as a function of hydrogen sulfide (H₂S and HS⁻) concentration at pH 6, 25°C, 500 bar, and the ionic strength of 0.5. Dissolved Fe²⁺ and Ni²⁺ concentrations in the early Archean seawater are estimated by Song et al. (2017) (10^{-2} – 10^{-4} M) and Konhauser et al. (2009; 2015) (4×10^{-7} M), respectively.

Response to Reviews

However as pointed out by this reviewer, there are still uncertainties on the availabilities of necessary reaction parameters, particularly on the methanethiol concentration, on the primordial seafloor. These geochemical availabilities must be investigated further to make a realistic estimate for the activity and robustness of abiotic thioester synthesis. We added this point in the revised manuscript (page 12 of Highlighted Revision):

“Thus, the thioester synthesis via the formation of Ni sulfide_PERMs should have been ~~robustly concomitant with ubiquitous hydrothermal activities on the primordial seafloor.~~ **possible in primordial ocean alkaline hydrothermal systems. Just as Huber and Wächtershäuser (1997), however, we have not yet discussed seriously the source of methanethiol. To make a realistic estimate for the activity and robustness of abiotic thioester synthesis, further investigation for the geochemical availabilities of thiols and the other necessary reaction parameters (the electric potential and the Ni-sulfide composition) is desirable.**”

(References)

- Bekker, A. et al. Atmospheric sulfur in Archean komatiite-hosted nickel deposits. *Science* **326**, 1086–1089 (2009).
- Konhauser, K. O. et al. Oceanic nickel depletion and a methanogen famine before the great oxidation event. *Nature* **458**, 750–753 (2009).
- Konhauser, K. O. et al. The Archean nickel famine revisited. *Astrobiology* **15**, 804–815 (2015).
- Shibuya, T., Russell, M. J. & Takai, K. Free energy distribution and hydrothermal mineral precipitation in Hadean submarine alkaline vent systems: Importance of iron redox reactions under anoxic conditions. *Geochim. Cosmochim. Acta* **175**, 1–19 (2016).
- Song, H. et al. (2017) The onset of widespread marine red beds and the evolution of ferruginous oceans. *Nat. Commun.* **8**, 399.

1-16. line 158 – I was wondering if there was any detection of thioacetate here.

RESPONSE

Response to Reviews

If thioacetate means a deprotonated form of thioacetic acid, we have not detected it. We only detected S-methyl thioacetate (MTA).

1-17. Figure 2a – the CO concentration with Fe(Ni)S is negligible at even -0.8 V here, despite the Fe:Ni ratio being 1:1. Even with -1 V, little CO forms with this most realistic of compositions (reflecting the Hadean ocean). This is why I suspect that if a 10:1 ratio of Fe:Ni had been used then CO concentration would have been undetectable. So I would say this is a proof of principle and does not reflect realistic Hadean conditions.

RESPONSE

The Ni/Fe ratio of sulfide deposits in primordial deep-sea vent environments may not have directly reflected the oceanic composition. See our response No. 1-15.

1-18. Fig 2b – again NiS works well but Fe(Ni)S much less well, despite a modest excess of methanethiol per sulfide. With Fe(Ni)S, yield is close to zero at -0.8 V, the generous Eh for Lost City today. All this ought to be acknowledged.

Fig 2c – here it seems as if Fe(Ni)S was successful as the percent conversion of CO to methyl thioacetate (MTA) is around 20-30 % even at lower potentials (-0.7 V) and this is reported (erroneously) in the abstract. But in fact it only appears to be good because there was so little CO reduced on this surface in the first place. This may give a helpful indication of the 2-step reaction mechanism but the straight claim seems misleading to me and ought to be modified.

RESPONSE

We specified in the revised manuscript that the yield of MTA in the presence of (Fe,Ni)S is low due to the low surface accumulation of CO during the electroreduction (page 10 of Highlighted Revision).

“Interestingly, even greater CO-to-MTA ~~reaction efficiencies~~ **conversion ratios** were obtained with the NiS coprecipitating with FeS or CoS (Figure 2c). Up to $56 \pm 10\%$ of the surface-bound CO, produced by CO₂ electroreduction, was converted to MTA on the electrolyzed (Co,Ni)S. **Although the electrolyzed (Fe,Ni)S produced MTA with low yields (Figure 2b) due to the low**

Response to Reviews

surface accumulation of CO during the electroreduction, the percentages of CO to form MTA were kept at high levels (~20%) even at ≥ -0.8 V_{SHE} (Figure 2c). In fact, (Fe,Ni)S produced a considerable amount of MTA (1.44 ± 0.29 μ mol) from initially externally introduced CO and methanethiol even after the -0.6 V_{SHE} electrolysis when -0.6 V_{SHE} was applied to the electrolysis (Figure 2d)."

For abstract, we added the percentage achieved by (Fe,Ni)S (see our response No. 1-4).

1-19. Fig 2d - with added CO. Here the pattern for Fe(Ni)S is surprising and implies that it is adsorbing onto other metal ions than simply native Ni (which reduces CO₂ to CO). This was not commented on much in the paper. Apart from anything else it implies that CO formed elsewhere may well be trapped within the system by adsorbing onto more surfaces.

RESPONSE

The observed excellent CO-to-MTA conversion on (Fe,Ni)S (Figure 2d) may not simply be due to the CO adsorption capability, given the multi-step MTA formation process (Figure 2e). Still, the role of (Fe,Ni)S in nature suggested by this reviewer is worth being mentioned in the manuscript. We added the following sentence on page 11 of Highlighted Revision.

"It is conceivable, for example, that the CO formed on NiS_PERM and CoS_PERM is trapped on (Fe,Ni)S_PERM with thiols, thereby realizing the thioester formation with mild electric potentials (Figure 2d)"

1-20. Figure legend – again one is entitled to wonder where all the methanethiol is coming from. It has been detected in hydrothermal systems but the consensus, insofar as there is one at all, is that it is derived from thermal decomposition of organic molecules buried deeper in the crust. So not available at the origin of life. This is contentious I know; but again the issue would be solved by presenting this work as proof of principle rather than 'solving' the problem. I would say it is closer to showing how the problem could be solved.

Response to Reviews

RESPONSE

As this reviewer points out, efficient mechanisms for methanethiol production on primordial seafloor remain unknown, although there is still a reaction space to be explored (see, for example, Kitadai et al., 2018). We have thus emphasized the necessity of future work in the revised manuscript (see our response No. 1-15). We also added a sentence about the methanethiol availability problem in the Figure 3 caption:

“Fig. 3 | Schematic cross-section of a vent chimney in ~~an early~~ a **primordial** ocean alkaline hydrothermal system showing possible abiotic thioester synthesis promoted by Ni sulfide_PERM. Note that the availability of methanethiol in primordial deep-sea vent environments remains controversial and needs further investigation (Heinen and Lauwers, 1996; Schulte and Rogers, 2004; Reeves et al., 2014).”

Still, we believe that the process demonstrated here for activating the CO₂-to-CO reduction and the CO-thiol reaction should have been a key for realizing the abiotic thioester synthesis. This possibility has been mentioned in the revised Abstract and Discussion. See our response No. 1-4 and 1-15.

(References)

- Heinen, W. & Lauwers, A. M. Organic sulfur compounds resulting from the interaction of iron sulfide, hydrogen sulfide and carbon dioxide in an anaerobic aqueous environment. *Orig. Life Evol. Biosph.* **26**, 131–150 (1996).
- Kitadai, N. et al. Geoelectrochemical CO production: Implications for the autotrophic origin of life. *Sci. Adv.* **4**, eaao7265 (2018).
- Reeves, E. P., McDermott, J. M. & Seewald, J. S. The origin of methanethiol in midocean ridge hydrothermal fluids. *Proc. Natl. Acad. Sci. USA* **111**, 5474–5479 (2014).
- Schulte, M. D. & Rogers, K. L. Thiols in hydrothermal solution: standard partial molal properties and their role in the organic geochemistry of hydrothermal environments. *Geochim. Cosmochim. Acta* **68**, 1087–1097 (2004).

Response to Reviews

1-21. Line 172 – I am not sure what the multiple numbers are referring to here, even when consulting Fig. S25. I assume w/wo externally added CO/MTA. If so, it is surprising to me that much more acetate than formate appears to be synthesized in the absence of both added CO and CH₃SH. That seems improbable so I suspect I am misunderstanding something. At least this needs to be made clear.

RESPONSE

We apologize for the confusion. Acetate was observed in appreciable amounts (>0.1 mM) only when methanethiol was added. The multiple numbers (for example, $0.19 \pm 0.04 \mu\text{mol}$ or $0.25 \pm 0.05 \text{ mM}$) were intended to present the amount ($0.19 \pm 0.04 \mu\text{mol}$) and concentration ($0.25 \pm 0.05 \text{ mM}$) of acetate. These expressions were modified in the revised manuscript, such as ($0.19 \pm 0.04 \mu\text{mol} = 0.25 \pm 0.05 \text{ mM}$ in 0.75 ml H₂O).

1-22. Line 187/88 – steep decline in yield of MTA below -0.8 V – which is borderline the values likely to be achieved in vents. It might be worth mentioning that the existence of awaruite in many fossil vent systems implies very high partial pressure of H₂, which would certainly lower the reduction potential to the required range (see e.g. Vasiliadou et al.).

RESPONSE

We appreciate the reviewer's suggestion. We mentioned awaruite as an indicator of the occurrence of elevated hydrothermal H₂ fluxes on the primitive seafloor (see our response No. 1-2).

1-23. Line 210 “(Fe,Ni)S produced a considerable amount of MTA from initially introduced CO and methanethiol even after the $-0.6 \text{ V}_{\text{SHE}}$ electrolysis (Figure 2d).” As noted above this is only because the amount of CO generated at -0.6 V was negligible. So a modest proportion of nearly nothing.

RESPONSE

The amount of surface-bound CO on (Fe,Ni)S is indeed low after the $-0.6 \text{ V}_{\text{SHE}}$ electrolysis, and thus CO has to be added externally to achieve a good MTA yield on this surface. We have discussed the importance of materials transport for abiotic thioester synthesis in the manuscript. See also our response No. 1-19.

Response to Reviews

1-24. Line 226 Potential level for Ni⁰ formation...depends a lot on how much Ni²⁺ was available relative to Fe²⁺; I think much less than implied here.

RESPONSE

The thermodynamic impact of Fe on the Ni²⁺/Ni⁰ redox potential ($\text{Ni}^{2+} + 2\text{e}^- \leftrightarrow \text{Ni}^0$) is difficult to be evaluated because of the lack of thermodynamic data for the bimetallic sulfides particularly in the amorphous state. But our observed greater performance of (Fe,Ni)S than NiS for promoting MTA synthesis at $-0.6 \text{ V}_{\text{SHE}}$ (Figure 2d) suggests that the Fe²⁺ coprecipitation with Ni²⁺ even facilitates the Ni⁰ formation. See also our response No. 1-15.

1-25. As an aside here it would also be good to clarify that ‘less than or equal to -0.6 V would mean -0.7 V , not -0.5 V . This is strictly correct as written but has potential to mislead, as “less than” might imply a less reducing (i.e. more positive) reduction potential. It would be useful to explicitly state that the use of less than or equal to refers to a more extreme reduction potential, more strongly negative, less achievable, whereas in the context it implies that the conditions could be even more mild than stated.

RESPONSE

“less” was originally used in the following sentence:

“Thus, NiS_PERM and CoS_PERM are formed at $-1.0 \text{ V}_{\text{SHE}}$ and even **less** negative potentials near their sulfide/metal equilibria (Figure 1b and f, Figures S11 and S12) just as in the FeS case.”

For clarity, we added a short annotation (i.e., closer to 0 V_{SHE}) after “less negative potentials”.

1-26. Line 236: “Thus, the thioester synthesis via the formation of Ni sulfide PERMs should have been robustly concomitant with ubiquitous hydrothermal activities on the primordial seafloor.” To

Response to Reviews

my mind this statement is too strong, given my comments above. I think the study proves the principle and the conditions required are quite mild and potentially overlap with those on the primordial seafloor. But the phrase at present is misleading: to form thioesters under these conditions requires very low reduction potentials (lower than -0.8 V, which is the lowest that has been detected in modern systems) coupled with high relative Ni^{2+} concentrations (or Co maybe), coupled with a high stoichiometric flux of methanethiol (which is highly implausible). So I think we would be deluding ourselves if we thought that this paper solves the whole problem, but it is important nonetheless because it shows what is possible. The conditions are right on the cusp of being realizable, and call for more work.

RESPONSE

We have modified this sentence and have emphasized the necessity of future works to solve the remaining problems. See our response No. 1-15.

1-27. Line 251 – Deep sea hydrothermal systems are referred to Fig. S1 but there is nothing in that figure on pressure. Pressure would of course increase the partial pressure of H_2 , lowering the reduction potential, making the conditions required more realistic. Again, this has not been clearly stated in the paper.

RESPONSE

We apologize for the lack of pressure information. 500 bar was considered here. This information was added in the figure caption:

“(right) Thermodynamic calculation for the H^+/H_2 and the CO_2/CO redox potentials (V_{SHE}) at 500 bar as a function of temperature and pH indicates that H_2 oxidation in hot and alkaline pH conditions readily generate negative electric potentials favorable for the CO_2 -to- CO conversion in cool (0–50 °C) and slightly acidic (pH 6–7) ancient seawater (Krissansen-Totton et al., 2018).”

The role of pressure in generating low electric potentials has been mentioned in the revised manuscript. See our response No. 1-2.

Response to Reviews

1-28. Fig S1 – I assume these calculations are based on the standard hydrogen electrode with 1 atmosphere pressure of H₂ (hence about 0.74 mg/kg dissolved H₂, well below those at Lost City). This could do with expanding.

RESPONSE

In this calculation, 1 mmol kg⁻¹ H₂ was considered because this is a typical H₂ concentration in fluids from the present-day serpentine-hosted hydrothermal systems (Schrenk et al., 2013; Tivey, 2007). A ten-fold change in the H₂ concentration changes the redox potential by ± ~30 mV at 25°C and by ± ~40 mV at 150°C. This explanation was originally made in Materials and Methods, but were transferred to the caption of Figure S1 in the revised manuscript.

“(right) Thermodynamic calculation for the H⁺/H₂ and the CO₂/CO redox potentials (V_{SHE}) at 500 bar as a function of temperature and pH indicates that H₂ oxidation in hot and alkaline pH conditions readily generate negative electric potentials favorable for the CO₂-to-CO conversion in cool (0–50 °C) and slightly acidic (pH 6–7) ancient seawater (Krissansen-Totton et al., 2018).

In this calculation, 1 mmol kg⁻¹ H₂ is considered because it is a typical H₂ concentration in fluids from the present-day serpentine-hosted hydrothermal systems (Tivey, 2007; Schrenk et al., 2013). The CO₂/CO activity ratio is set to one. Equilibrium calculation with this ratio gives the potential conditions where CO₂ and CO are equally stable. A ten-fold change in the molecular species concentration changes the redox potential by ± ~30 mV at 25°C and by ± ~40 mV at 150°C.”

(References)

- Krissansen-Totton, J., Arney, G. N. & Catling, D. C. Constraining the climate and ocean pH of the early Earth with a geological carbon cycle model. *Proc. Natl. Acad. Sci. USA* **115**, 4105–4110 (2018).
- Schrenk, M. O., Brazelton, W. J. & Lang, S. Q. Serpentinization, carbon, and deep life. *Rev. Mineral. Geochem.* **75**, 575–606 (2013).
- Tivey, M. K. Generation of seafloor hydrothermal vent fluids and associated mineral deposits. *Oceanography* **20**, 50–65 (2007).

Response to Reviews

1-29. I won't comment more on the SI figures, except to call again for a little more detail on the XANES derivatizations. I would like to say though that there are really beautiful data, a joy to behold how well this work has been done. The team are to be congratulated on the quality of their data.

That's all. I hope my comments are taken in the spirit they are intended, to improve (a little) an excellent paper that clearly deserves publication in Commun Chem, and should excite a wide readership in answering a long-standing, difficult and important question. It out to be highly cited.

I am happy for the authors to know that I am

Nick Lane

RESPONSE

We thank the reviewer's kind recommendation again.

Response to Reviews

Responses to the Reviewer 2's comments

2-1. The manuscript entitled 'Thioester synthesis through geoelectrochemical CO₂ fixation on Ni sulfides' by Kitadai et al. presents results about the catalytic properties of the NiS/Ni(0) system to reduce CO₂ to CO and finally the reaction with thiols to corresponding thioesters, here S-methyl thioacetate. This is an interesting process, which might be a puzzle piece in the emergence of life or at least an early step to the acetyl-CoA pathway, which is the energy-releasing route of biological CO₂ fixation. This adds to already known and well described properties of Fe and Ni sulfides (Wächtershäuser and co-authors), which were extensively studied in the past as catalyst. The new findings are interesting in particular in the context of the observed spontaneous generation of electricity in deep-sea vent chimneys and mineral deposits (Yamamoto et al.). Another important aspect is that there is still a gap of knowledge in the explanation of the thioester-dependent acetyl-CoA metabolism. In general, there are many and detailed aspects in the context of the emergence of life, which are of interest for the broader community. The experiments might trigger further investigations on reactions, where there sub stoichiometric sulfides may play a role as catalysts.

RESPONSE

We are grateful to Reviewer 2 for his/her very favorable evaluation.

2-2. A critical point is the aging of the NiS samples in the partially reduced state. The authors should comment on the long-term stability of the synthesized particles and the change of the catalytic properties.

RESPONSE

In a glove box filled with a H₂-N₂ mixed gas (the volumetric H₂/N₂ ratio = 4/96), the sulfide_PERMs can be stored over weeks without degrading the redox states and catalytic performances, but they are oxidized almost completely in the air within minutes. This information was added in the revised Materials and Methods (page 3 of Supplementary Information):

"The prepared sulfide_PERMs were stable in the anaerobic glove box over weeks, but were rapidly oxidized in the air. Typically, a minute exposure to the air completely degraded their capabilities for promoting MTA synthesis. Their high susceptibilities to oxidation are not a

Response to Reviews

severe problem in the geoelectrochemical scenario (Figure S1) because of the absence of O₂ and other reactive oxidants (for example, H₂O₂) in primordial deep-sea environments.”

2-3. For the curiosity, is there a size-dependency observed in the XANES measurements?

RESPONSE

Besides the valence state, the X-ray absorption spectrum particularly in the EXAFS region reflects the local environment of a target element. To minimize the influence of particle size, we analyzed the XANES region (8300–8440 eV for NiS and 7690–7830 eV for CoS) for estimating the percentages of zerovalent metal in the electrolyzed sulfides. See the XANES data analytical method added in response to the 12th comment of Reviewer 1.

2-4. I expect that there are phase transitions in the NiS upon reduction. Is there any evidence for phase transition, which could cause the formation of highly active sites?

RESPONSE

We have interpreted from the XRD, SEM-EDS, and XANES data that the Ni⁰ in NiS_PERM are finely dispersed (Figure 1). The microscopic Ni⁰ clusters might be considered as a result of phase transition crucial for enhancing the MTA formation, because pure NiS, Ni₃S₂, and Ni⁰ did not promote (or very poorly promoted) the desired reaction (Figure S21, Table S4).

2-5. In the experimental section of the supplementary information there is a comment about the glovebox that 4% of hydrogen gas is used to avoid oxidation. It is important to prove that the prepared particles do not have absorbed hydrogen. I understand that the experimental prove and potential desorption might be difficult, but I would recommend the use of D₂O in the CO conversion to S-methylthioacetate.

RESPONSE

Response to Reviews

The MTA synthesis was examined in a serum bottle after flushing out the 4% H₂-containing gas with pure CO₂ gas (see Materials and Methods). Although H₂ was observed after the experiments (Table S3 and S4), the amount increased with decreasing the potential applied to the sulfide electroreduction, indicating that this H₂ derived from the proton reduction by the sulfide_PERMs ($2\text{H}^+ + 2\text{e}^- \rightarrow \text{H}_2$), rather than the H₂ adsorbed on the surfaces in the glovebox. In either case, it is unlikely that H₂ serves as a hydrogen source for the MTA synthesis because hydrogens in MTA are only at the methyl group. To the best of our knowledge, there is no experimental demonstration of a methyl synthesis from CO (or CO₂) and H₂ under mild aqueous condition. In fact, we did not observe any methyl-containing compounds including acetate, methane, and MTA with appreciable amounts in the absence of CH₃SH (Figure S25, Table S3 and S4).

2-6. Please explain in more detail the mentioned CO-to-MTA reaction efficiencies because this is only a qualitative measure. If possible, given numbers, in which way the ‘efficiency’ was improved. Same holds for the ‘considerable amount of MTA’ in line 210.

RESPONSE

For clarity, “CO-to-MTA reaction efficiencies” was replaced with “CO-to-MTA conversion ratios” We also added the yield of MTA soon after the “considerable amount of MTA”. See our response No. 1-18, where the modified sentences are presented.

2-7. In lines 211 and 212 there is reference given to Figures 7 and 8. Please add in the supplementary information.

RESPONSE

We corrected our oversight: “Figures 7 and 8” was replaced with “Figures S7 and S8”.

Response to Reviews

2-8. A minor point, which requires some better definition and chemical description is the term FeS_PERM for the FeS, partially electro reduced to metal. I think this could be more precisely expressed in a formula e.g. Fe_xS , defining $x > 1$ or $\text{Fe}_{>1}\text{S}$, where x and >1 are subscript.

RESPONSE

We thank the reviewer's suggestion. But since the chemical formulas with $x > 1$ also represent some reduced sulfides (for example, Ni_3S_2 and Co_9S_8), such a formula may not clearly indicate the presence of zerovalent metals. Quantification of the metal valence states in every sample is difficult especially for the bimetallic sulfides. Thus, at present, we would like to keep the use of the term "sulfide_PERM".

Response to Reviews

Responses to the Reivewer 3's comments

3-1. It is a beautiful work and an interesting hypothesis as an effort to understand the electrochemical reactions in the early earth. The experimental design is well-established and this work can provide many insights for the future studies. For the publication, the following points should be improved and addressed.

RESPONSE

We appreciate the reviewer's favorable evaluation and valuable comments/suggestions listed below.

3-2. The results are fascinating and contains a lot of new progress but the logical flow in the manuscript is a little bit confusing. In fig 1 and fig 2, the electrochemical analysis is done for all meatal sulfides. We understand that it is a kind of the screening to identify the excellent activity of NiS. But for the readers, the flow seems not focused. As the author wanted to emphasize the Ni, it would be better to move some data to supporting information.

RESPONSE

As Reviewer 1 commented several times (see No. 1-15, 1-17, 1-18, 1-19, 1-23, and 1-24), the performances of sulfides other than NiS, particularly FeS, are also important topic in this study given the metal availability on the primitive seafloor. On the other hand, because of the space limitation of Communications Chemistry article (for example, a maximum of 5 display items is allowed), it is difficult to present all solid analytical data in the main text. Thus, we would like to keep the present Figure 1 and 2.

To make what follows clearer, we added the following sentence at the end of Introduction (page 4 of Highlighted Revision):

“In the following and in Supplementary Information, we also present the results for FeS and CoS and their influences on the NiS's capabilities for CO production and MTA synthesis.”

Response to Reviews

3-3. In this regard, the reason why Ni containing sulfide is better than Co, Fe sulfide is not clear. Also, it was mentioned that NiS with Co and Fe is the best. But the scientific discussion should be added in the details.

RESPONSE

Elucidation of the reaction mechanism is indeed desirable. The difficulty arises, however, from the fact that the efficient MTA formation occurred on NiS_PERM while pure NiS, Ni₃S₂, and Ni⁰ did not promote (or very poorly promoted) the reaction (Table S3 and S4). Thus, knowledge about pure solid–methanethiol (or CO) interactions, which is available in the literature, do not provide the reason for the exceptional capability of NiS_PERM. At present, our best interpretation based on our experimental results is the one given at Figure 2e. Further clear and definitive explanation requires detailed computational and spectroscopic investigations of the interfacial processes. Also note that the experimental demonstration of thioester synthesis is never a trivial work: there is no theoretical work that predicted or even anticipated the condition suitable for this reaction, even though its significance in protometabolism has been suggested over several decades (see Reviewer 1's comment No. 1-1). We thank the reviewer for this important comment. We would like to tackle this problem in a future work.

3-4. All the Ni on the surface of NiS during the electrolysis can be bound with CO? Quantitative analysis can be necessary to get the number of active site of Ni for the CO attachment.

RESPONSE

Given the crystal structure of heazlewoodite (Fleet, 1977; Figure S13), two Ni atoms are expected to be present on a unit cell surface area of 16.66 Å², which corresponds to 2.0 × 10⁻⁵ mole of surface Ni atoms per m². If we assume the electrolyzed NiS particle to be a sphere of diameter 17 nm on the average (Figure S6), 180 ± 40 μmol g⁻¹ of the adsorbed CO (Figure 2a) corresponds to one CO molecule per 8 surface Ni atoms. Thus, assuming an even distribution of 12% Ni⁰ in the NiS_PERM structure, the surface Ni⁰ sites are expected to be largely bound with CO. See Figure S13 and the figure caption.

(References)

Response to Reviews

Fleet, M. E. The crystal structure of heazlewoodite, and metallic bonds in sulfide minerals. *Am. Mineral.* **62**, 341–345 (1977).

3-5. If the methanethiol exist together during the CO₂-CO electrochemical conversion, the yield to thioester can be decreased a lot? Why is the separate step necessary?

RESPONSE

We did not carry out “one-pot” MTA synthesis in an electrochemical cell because H₂ evolution during the sulfide electrolysis prevented us from keeping methanethiol in the cell in the reaction period due to pressurization (see Materials and Methods, where we have made this explanation). If an open system is used with methanethiol being supplied continuously during the electrolysis, the MTA synthesis rate is expected to increase compared with the two-step reaction because of the continuous CO production and the activation of the sulfide_PERM surface by desulfurization. Such materials transport and combination of multiple reactions may have been rather realistic within the pore spaces of ancient hydrothermal mineral deposits as discussed in our manuscript (see page 12 of Highlighted Revision and Figure 3).

3-6. In product analysis, some organic molecules such as pyruvate were detected. But the mechanism is not clear.

RESPONSE

We did not observe pyruvate in any product solutions unless the sample solutions were exposed to the air for several days after the basification. The resultant micromolar level pyruvate is a contaminant rather than a product (please see Materials and Methods). Pyruvate synthesis with micromolar concentrations was recently reported by Varma et al. (2018) and Preiner et al. (2020). Please see their reports, where a possible mechanism of pyruvate formation has been given.

(References)

Response to Reviews

Varma, S. J., Muchowska, K. B., Chatelain, P. & Moran, J. Native iron reduces CO₂ to intermediates and end-products of the acetyl-CoA pathway. *Nat Eco Evol* **2**, 1019–1024 (2018).

Preiner, M. et al. A hydrogen-dependent geochemical analogue of primordial carbon and energy metabolism. *Nat Eco Evol* **4**, 534–542 (2020).

3-7. In the mantle, there is also FeS. The competition between Fe and Ni for the CO₂/CO binding will affect or we can consider the cooperative mechanism.

RESPONSE

We have examined the influences of Fe on the CO production and the MTA synthesis on NiS_PERM. See Figure 2.

3-8. Is the concentration gradient of sulfur and hydrogen enough to drive the electrochemical reduction of CO₂ to C?

RESPONSE

The CO₂/C redox potential ($\text{CO}_2 + 4\text{H}^+ + 4\text{e}^- \leftrightarrow \text{C} + 2\text{H}_2\text{O}$) at pH 6, 25°C, and 500 bar in the presence of 30 mM CO₂ is $-0.15 \text{ V}_{\text{SHE}}$, which is attainable in many natural hydrothermal systems (see Fig. 2 of Boyd et al., 2020). In fact, graphite has been found in deep-sea vent environments (Estes et al., 2019). However, such natural graphite is likely formed through hydrothermal processes beneath the ocean floor. As far as we know, there is no experimental demonstration of the electrochemical C production from CO₂.

(References)

Boyd, E. S., Amenabar, M. J., Poudel, S. & Templeton, A. S. Bioenergetic constraints on the origin of autotrophic metabolism. *Phil. Trans. R. Soc. A* **378**, 20190151 (2019).

Estes, E. R. et al. Abiotic synthesis of graphite in hydrothermal vents. *Nat. Commun.* **10**, 5197 (2019).

Response to Reviews

3-9. Ni or transition metals can act as a catalyst for the reaction of CO and methanethiol. Is it unexpected?

RESPONSE

There are several pieces of information suggesting the NiS_PERM's capability for promoting the CO-methanethiol reaction, including (1) Huber and Wächtershäuser's demonstration of MTA synthesis in a low yield using pure NiS, (2) the Ni's redox state change between +1 and +3 in the acetyl-CoA synthetase catalytic cycle (Can et al., 2017), and (3) the electrochemical formation of FeS_PERM and its excellent performance for facilitating several organic reactions (Kitadai et al., 2019). In contrast, identification of the NiS_PERM formation requires close EDS and XANES analyses (Figure 1). Thus, we did not start this study without some relevant expectations, but the effectiveness and crucial necessity of Ni-sulfide_PERMs were not clearly anticipated.

(References)

- Can, M., Giles, L. J., Ragsdale, S. W. & Sarangi, R. X-ray absorption spectroscopy reveals an organometallic Ni–C bond in the CO-treated form of acetyl-CoA synthase, *Biochem.* **56**, 1248–1260 (2017).
- Huber, C. & Wächtershäuser, G. Activated acetic acid by carbon fixation on (Fe,Ni)S under primordial conditions. *Science* **276**, 245–247 (1997).
- Kitadai, N. et al. Metals likely promoted protometabolism in early ocean alkaline hydrothermal systems. *Sci. Adv.* **5**, eeav7848 (2019).

Reviewers' comments:

Reviewer #1 (Remarks to the Author):

I have been over the authors' response to my comments and revisions to the MS and I am happy that they have made entirely suitable changes throughout. I reiterate that this is an important paper with beautiful data, and I believe it now has a good balance in referring to a wider and older literature. I am disappointed that the authors' failed to replicate Roldan et al. I would be genuinely surprised if that work had not been well done, but I accept the authors' explanation for not citing it. Finally I thank for authors' for taking my comments in such a constructive spirit (as I had hoped), and for their kind acknowledgement. I strongly recommend publication in Comms Chem.
Nick Lane

Reviewer #2 (Remarks to the Author):

The authors answered all open questions and performed all requested corrections and changes. This is important research about one of the key transformations leading to the emergence of life that deserves to be published without further changes.

Reviewer #3 (Remarks to the Author):

Decision: Accept with minor revision

Prebiotic CO₂ fixation is considered as the most fundamental steps for the origin of life, because geochemical CO₂ fixation is the pondered common phenomenon on terrestrial planets due to the widespread hydrothermal activity in our solar system. In such a context, the demonstration of the formation of thioesters under simulated geoelectrochemical condition in this work brands this manuscript is more suitable for the publication. Besides, thioesters were considered as the precursors to pyrophosphate, which in turn produced ATP, a molecule that is used to supply energy for many of life's processes today. The following points may be addressed before publication.

1. Page 10; line 243: In the revised manuscript it was mentioned that, the yield of MTA in the presence of (Fe,Ni)S is low due to the low surface accumulation of CO during the electroreduction. If I am correct, iron can strongly coordinate with CO rather than Ni, but the explanation leading that iron incorporation reduced the CO accumulation. The author should address this issue more clearly.

2. The formation of Ni⁰ and the mixed valence state of NiS facilitated the MTA formation because of the strong binding affinity of Ni⁰ rather than Ni^{II}. Therefore, the incorporation of iron enhancing the MTA formation might be related to the CO binding affinity of iron rather than other kind of mechanism. Can the author explain this point more clearly?

In overall, this work is more suitable for publication after minor revisions.

Responses to the Reviewer 3's comments

1-1. Prebiotic CO₂ fixation is considered as the most fundamental steps for the origin of life, because geochemical CO₂ fixation is the pondered common phenomenon on terrestrial planets due to the widespread hydrothermal activity in our solar system. In such a context, the demonstration of the formation of thioesters under simulated geoelectrochemical condition in this work brands this manuscript is more suitable for the publication. Besides, thioesters were considered as the precursors to pyrophosphate, which in turn produced ATP, a molecule that is used to supply energy for many of life's processes today.

RESPONSE

We are grateful to Reviewer 3 for his/her additional valuable comments. Our manuscript was indeed improved significantly owing to the first round of peer review by this and the other two reviewers.

In the Abstract revised according to the reviewers' comments, we have stated that thioester formation was demonstrated under a simulated geoelectrochemical condition (page 2). In addition, the possibility that thioesters might have preceded the phosphorus-based energy currencies (e.g., ATP) has been mentioned in the revised Introduction (page 3).

1-2. The following points may be addressed before publication.

Page 10; line 243: In the revised manuscript it was mentioned that, the yield of MTA in the presence of (Fe,Ni)S is low due to the low surface accumulation of CO during the electroreduction. If I am correct, iron can strongly coordinate with CO rather than Ni, but the explanation leading that iron incorporation reduced the CO accumulation. The author should address this issue more clearly.

RESPONSE

Although Fe⁰ has slightly stronger CO binding affinity than Ni⁰ (Abild-Pedersen & Andersson, 2007; this reference has been cited in page 5 of supplementary information), the CO binding strength does not directly determine the metal's catalytic capability for CO₂-to-CO electroreduction (Hansen et al., 2013). In fact, no CO production on FeS_PERM was observed (Figure 2a; Kitadai et al., 2018). On the other hand, no CO evolution occurred during the NiS electrolysis (Kitadai et al., 2018), indicating that the CO–Ni⁰ bonds on

NiS_PERM are stable in the examined electrochemical condition. Thus, the surface Fe sites on (Fe,Ni)S_PERM is likely not involved in the CO production and accumulation.

We made this point clear in the revised manuscript as follows (page 6):

“Coprecipitation with FeS led to decline of the amounts of CO on NiS and CoS (Figure 2a) probably ~~because of the decrease in the surface reactive sites~~ because the resultant surface Fe sites were not involved in the CO accumulation owing to the stable Ni⁰-CO and Co⁰-CO bindings, as indicated by no CO evolution during the NiS and CoS electrolysis (Kitadai et al., 2018).”

(References)

Abild-Pedersen, F. & Andersson, M. P. CO adsorption energies on metals with correction for high coordination adsorption sites – A density functional study. *Surf. Sci.* **601**, 1747–1753 (2007).

Hansen, H. A., Varley, J. B., Peterson, A. A. & Norskov, J. K. Understanding trends in the electrochemical activity of metals and enzymes for CO₂ reduction to CO. *J. Phys. Chem. Lett.* **4**, 388–392 (2013).

Kitadai, N. et al. Geoelectrochemical CO production: Implications of the autotrophic origin of life. *Sci. Adv.* **4**, ea07265 (2018).

1-3. The formation of Ni⁰ and the mixed valence state of NiS facilitated the MTA formation because of the strong binding affinity of Ni⁰ rather than Ni^{II}. Therefore, the incorporation of iron enhancing the MTA formation might be related to the CO binding affinity of iron rather than other kind of mechanism. Can the author explain this point more clearly?

In overall, this work is more suitable for publication after minor revisions.

RESPONSE

We thank the reviewer for this suggestion. But the observed CO-to-MTA conversion on (Fe,Ni)S (Figure 2d) may not simply be due to the CO binding affinity, given the multi-step MTA formation process (Figure 2e). Note also that pure FeS, NiS, Ni₃S₂, Fe⁰ and Ni⁰ did not promote (or very poorly promoted) the reaction whereas the efficient MTA formation occurred on (Fe,Ni)S_PERM and NiS_PERM (Table S3 and S4). Thus, the knowledge about

pure solid–CO (or methanethiol) interactions may not directly explain the reaction mechanism. The mechanistic elucidation would require detailed computational and spectroscopic analyses of the interfacial processes, which are beyond the scope of this study; the present chief aim is to report a previously unknown chemical reaction route crucial for the origin of life.

REVIEWERS' COMMENTS:

Reviewer #3 (Remarks to the Author):

All the concerns were clarified. Now, it can be published.